# Disentangling Superpositions: Interpretable Brain Encoding Model with Sparse Concept Atoms

**Alicia Zeng**
Biophysics Program
University of California, Berkeley
Berkeley, CA 94720
litrtwe@berkeley.edu

**Jack Gallant**
Department of Neuroscience
University of California, Berkeley
Berkeley, CA 94720
gallant@berkeley.edu

## Abstract

Encoding models using word embeddings or artificial neural network (ANN) features reliably predict brain responses to naturalistic stimuli, yet interpreting these models remains challenging. A central limitation is superposition: distinct semantic features become entangled along correlated directions in dense embeddings when latent features outnumber embedding dimensions. This entanglement renders regression weights non-identifiable—different combinations of semantic directions can produce identical predictions, precluding principled interpretation of voxel selectivity. To address this, we introduce the Sparse Concept Encoding Model, which transforms dense embeddings into a higher-dimensional, sparse, non-negative space of learned concept atoms. This transformation yields an axis-aligned semantic basis where each dimension corresponds to an interpretable concept, enabling direct readout of conceptual selectivity from voxel weights. When applied to fMRI data collected during story listening, our model matches the prediction performance of conventional dense models while substantially enhancing interpretability. It enables novel neuroscientific analyses such as disentangling overlapping cortical representations of time, space, and number, and revealing structured similarity among distributed conceptual maps. This framework offers a scalable and interpretable bridge between ANN-derived features and human conceptual representations in the brain.

## 1 Introduction

Artificial neural networks (ANNs) were originally inspired by the brain, yet progress in machine learning has far outpaced advances in understanding brain function. A central bottleneck in neuroscience is data: while many large-scale datasets are available for training machine learning models, brain recordings remain scarce and costly. To overcome this limitation, neuroscientists increasingly use machine learning models to study how the brain represents information. A common framework for this approach is the **encoding model**, which predicts brain activity from quantitative stimulus features using linear regression [22, 27, 15, 35, 56]. Early encoding models relied on manually-defined features, such as labeled objects or actions in movies [23, 55, 34]. Recent encoding models instead leverage features derived from machine learning, such as word embeddings or activations from large language models (LLMs) [52, 30, 37, 57, 1, 24, 20, 28, 50, 26]. When used as regressors to predict brain responses to naturalistic stimuli such as narrative stories, machine-learning-derived features consistently outperform manually-defined features. In particular, features from deeper neural networks achieve higher predictive accuracy [1, 7, 45].

Despite their success in applications like speech decoding [29, 51, 47], ANN-based encoding models remain limited in their ability to advance fundamental neuroscience due to poor interpretability.

39th Conference on Neural Information Processing Systems (NeurIPS 2025).

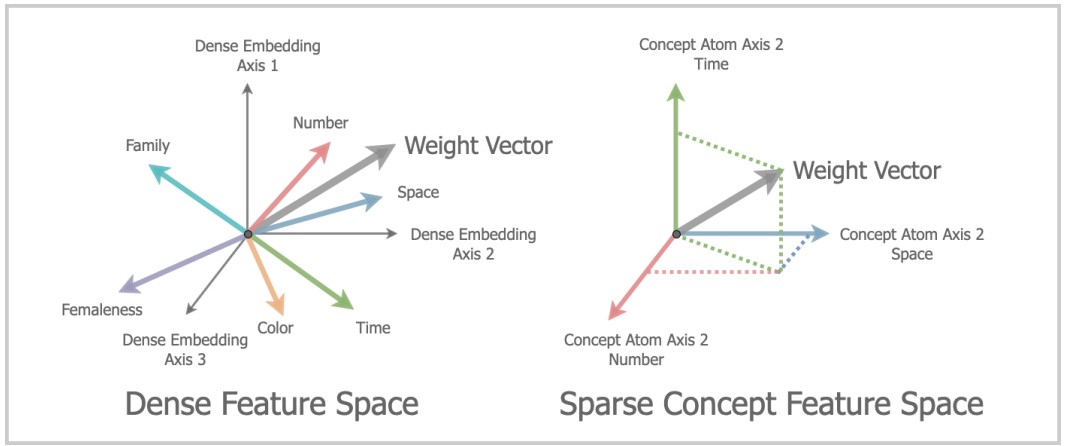

Figure 1: **The Sparse Concept Encoding Model resolves superposition in dense feature spaces.**
**Left:** In dense embedding space, semantic concepts are represented by correlated linear combinations
of basis directions. This superposition introduces ambiguity: a voxel's weight vector may project
positively onto multiple concept directions simply because those directions are themselves correlated.
**Right:** The Sparse Concept Encoding Model recovers the underlying semantic directions and
assigns each to an independent axis, transforming the dense space into an interpretable, axis-aligned
representation. This enables voxel tuning to be directly read out along concept dimensions.

Mapping ANN-derived representations onto brain activity often amounts to explaining one black
box with another, yielding little insight into the underlying neural mechanisms. However, many core
questions in neuroscience, such as where and how abstract concepts are represented in the brain,
demand models with interpretable internal structure.

In this work, we address the interpretability challenges inherent to dense feature representations in
encoding models. Previous studies using word embeddings or ANN-based features often assume that
regression weights in these high-dimensional spaces are directly interpretable [22, 50, 54, 13]. We
show that this assumption is fundamentally flawed. Specifically, we demonstrate that *superposition*,
the entanglement of multiple latent semantic features when their number exceeds the embedding
dimensionality, introduces intrinsic ambiguity in model interpretation [16, 8]. When semantic features
are superposed, regression weights become **non-identifiable**: distinct combinations of semantic
features can produce the same predicted brain responses. This makes it impossible to reliably identify
which specific concepts a voxel encodes, limiting the utility of dense models for neuroscientific
inquiries. To overcome this limitation, we propose a new framework that preserves the predictive
power of dense features while significantly enhancing model interpretability. Our method disentangles
superposed representations into sparse, semantically-aligned concept dimensions, allowing a clearer
link between model structure and brain activity. The framework generalizes naturally across a
wide range of representations, such as LLM embeddings, and across recording modalities, such as
electrocorticography (ECoG) and magnetoencephalography (MEG).

**Main Contributions**    (1) We show that regression weights learned from dense features are inherently
non-identifiable. Using a Bayesian analysis, we trace this ambiguity to the implicit prior induced by
the geometry of dense embedding spaces. (2) We introduce the Sparse Concept Encoding Model,
which transforms dense features into a higher-dimensional, sparse space composed of learned concept
atoms. (3) We propose Vector Norm Reparameterization (VNR), a preprocessing method that
enhances interpretability of learned concept atoms by expanding the relative spread of low-norm
vectors while compressing the range of high-norm vectors. (4) We apply the Sparse Concept Encoding
Model to a naturalistic fMRI dataset and demonstrate two neuroscientific analyses that are infeasible
with dense models: (i) disentangling overlapping cortical representations of time, space, and number,
and (ii) comparing concept representations across the cortex to reveal structured relationships among
distributed conceptual maps.

| 3 ✖ | Concept Atom 361 "Trail": hilly, uphill, jog, trail, mile |
| + 2 ✖ | Concept Atom 924 "Walkability": rambla, subway, boulevard, avenue, street |
| Walking = + 2 ✖ | Concept Atom 384 "Present Continuous Tense": accusing, owning, criticizing, wasting |
| + 2 ✖ | Concept Atom 456 "Body position": sitting, crouching, kneeling, crouched, standing |
| + 1 ✖ | Concept Atom 997 "Space": centre, near, situated, located, km |

Figure 2: **Sparse dictionary learning decomposes each word into a sparse, non-negative linear combination of concept atoms.** This example shows the sparse decomposition of the word "walking" using the learned concept atom dictionary. Each contributing atom is shown with its index, a descriptive label (chosen heuristically by the authors), and its top 5 activating stimulus words from the story transcripts.

## 2 Theories

### 2.1 Sparse Coding

Sparse coding, also known as sparse dictionary learning, was originally proposed as a model for how the visual cortex efficiently represents natural scenes [38, 39]. The central idea is that many natural signals, such as images, sounds, or words, can be expressed as linear combinations of a small number of latent components, often called *basis factors* or *atoms*. For example, the word `queen` can be decomposed into two latent factors: `royalty` and `femaleness`. These features tend to occur independently: "royalty" appears in "king" and "castle", while "femaleness" is present in "daughter" and "mother". Since most words activate only a small number of such factors, the resulting representations are **sparse**.

Formally, each input $\mathbf{x}_j \in \mathbb{R}^p$ is approximated as a linear combination of atoms from an overcomplete dictionary $\{\phi_i\}_{i=1}^m \subset \mathbb{R}^p$, where $m > p$: $\mathbf{x}_j = \sum_{i=1}^m a_{ij}\phi_i$. The coefficient vector $\mathbf{a}_j \in \mathbb{R}^m$ is constrained to be sparse, with most entries equal to zero. Overcompleteness ($m > p$) allows the dictionary to capture fine-grained structure in the data but renders the linear system underdetermined. Sparsity resolves this ambiguity by favoring solutions that use only a small subset of atoms. Specifically, the dictionary and coefficients are jointly learned by minimizing a regularized least-squares objective:

$$\min_{\{\mathbf{a}_j\},\{\phi_i\}} \sum_{j=1}^n \left\| \mathbf{x}_j - \sum_{i=1}^m a_{ij}\phi_i \right\|_2^2 + \lambda \sum_{j=1}^n \|\mathbf{a}_j\|_1,$$

### 2.2 Sparse Coding For Interpreting Word Embeddings and Deep Neural Networks

Word embeddings are known to encode semantic features as linear directions in activation spaces. The canonical example by Mikolov et al. [32], `king` − `man` + `woman` ≈ `queen`, showed that vector arithmetic on word embeddings captures analogies. This suggests that certain linear directions correspond to interpretable semantic concepts (`royalty` and `femaleness`). Arora et al. (2016) [2] provided a theoretical explanation for the linear structure through a *random-walk-on-discourses* model, where word vectors are generated from mixtures of latent discourse vectors. Arora et al. (2018) [3] further argued that sparse coding provides a natural method to recover these latent discourse directions from dense embeddings [17]. Subsequent studies applying sparse coding to word embeddings have indeed revealed dictionary atoms that correspond to interpretable semantic and syntactic features [3, 4, 59].

Recently, sparse coding has been adopted in mechanistic interpretability research on deep neural networks (DNNs), which aims to decompose models into human-interpretable components [43, 46]. Early efforts focused on analyzing individual neurons, under the assumption that each neuron corresponds to a single interpretable feature [25, 36, 5]. However, empirical studies revealed that many neurons respond to multiple, often unrelated features, a phenomenon known as *polysemanticity* [36]. To explain this, researchers proposed the *superposition hypothesis* [16, 8]: when the number of latent features exceeds the number of available neurons, the network encodes each feature as a distinct

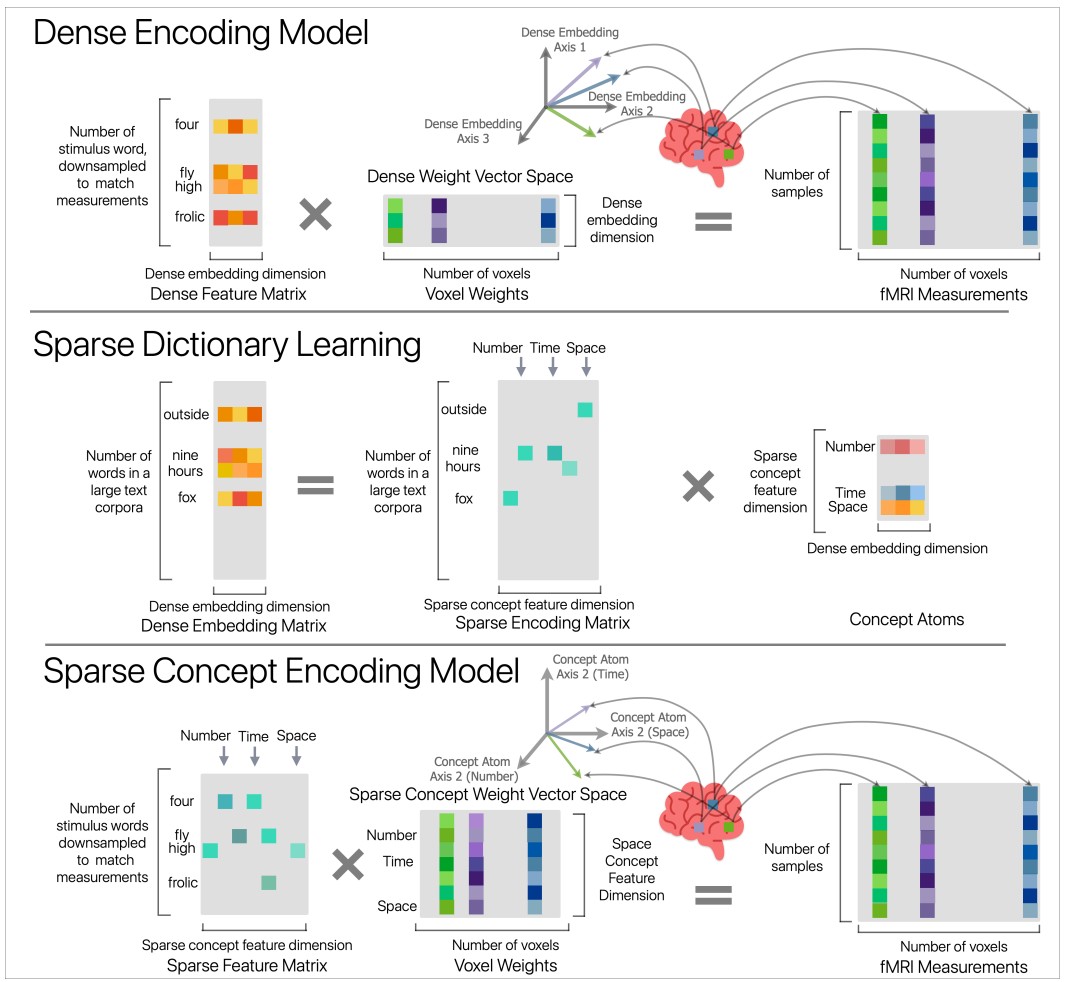

Figure 3: **The Sparse Concept Encoding Model produces interpretable voxel weights by replacing dense features with learned concept atoms. Top:** *Dense Encoding Model*: Stimuli are represented using dense word embeddings, resulting in voxel weights that lie in an entangled feature space where semantic directions overlap. **Middle:** *Sparse Dictionary Learning*: Dense embeddings are transformed into a higher-dimensional space of sparse concept atoms, each defining a distinct, interpretable semantic axis. **Bottom:** *Sparse Concept Encoding Model*: Stimuli are re-encoded as sparse combinations of concept atoms. The resulting voxel weights are defined with respect to disentangled concept-atom axes, enabling direct and interpretable readout of semantic selectivity.

linear combination of neurons. Superposition thus emerges naturally as an efficient representational strategy under data sparsity and limited capacity. Inspired by earlier work on word embeddings, recent studies have applied sparse coding to dense DNN activations to recover interpretable basis factors [10, 58, 21]. In particular, sparse autoencoders trained on large language model (LLM) activations have shown that a few hundred neurons can represent thousands of distinct, interpretable features in superposition [8].

## 2.3   Limitations of Dense Encoding Models

The seminal work of Huth et al. [22] used word embeddings to represent narrative stimuli and revealed widespread conceptual representations across the human cortex. This approach catalyzed a growing body of research using machine-learning-derived features to predict brain activity. These studies typically employ regularized linear regression to fit a separate model for each voxel (a volumetric unit of brain tissue measuring approximately $3 \times 3 \times 4$ millimeters), producing a weight vector that characterizes the voxel's feature selectivity [15, 35, 30].

However, interpreting regression weights from dense encoding models presents a fundamental ambiguity. In dense embeddings (Fig. 1, left), the coordinate axes are not inherently interpretable; semantic concepts are represented by correlated linear combinations of basis directions. This entangled structure complicates interpretation, as the relationship between voxel weights and semantic concepts becomes opaque. Conventional analyses often rely on heuristic assumptions: for example, inferring that a voxel is selective for a concept if its weight vector projects positively onto that concept's direction. But such inferences are unreliable when concept directions are correlated. A voxel aligned with *number*, for instance, may appear to also respond to *time* simply because the two concept directions are positively correlated. More problematically, many concept pairs, such as *femaleness* and *space* in the diagram, may be strongly anti-correlated, making it unclear whether a voxel is activated by one, suppressed by the other, or both. These ambiguities undermine principled interpretation: a positive projection does not necessarily imply true selectivity to a concept; observed overlaps in cortical maps may reflect either genuine multi-concept selectivity or spurious effects induced by embedding geometry. The root of this ambiguity is formalized in the **Theorem of Non-Identifiability in Dense Embedding Spaces** (Appendix B). In contrast, an ideal feature space would have an interpretable basis in which each axis corresponds to a distinct semantic concept (Fig. 1, right). In such a space, a voxel's tuning could be directly read from its projections along concept axes. This axis-aligned structure is the interpretability goal we aim to achieve with the Sparse Concept Encoding Model.

## 2.4  The Sparse Concept Encoding Model

To circumvent the identifiability limitations of dense embeddings, we aim to recover the latent semantic directions and assign each its own coordinate, effectively lifting the dense feature space into an interpretable, axis-aligned representation (Fig. 1, right). We achieve this through non-negative sparse dictionary learning applied to a large text corpus, producing an overcomplete set of semantic axes that we term **concept atoms**. Specifically, we expand the original $d$-dimensional embedding space into an $m$-dimensional space ($m > d$), where each axis corresponds to a distinct concept atom (Fig. 3, middle). Each stimulus vector is then re-encoded as a sparse, non-negative linear combination of these atoms (see Fig. 2 for an example). This transformed representation replaces the original dense features in the encoding model. In the resulting sparse feature space, voxel weights become directly interpretable: each dimension aligns with a distinct semantic concept, enabling principled readout of conceptual selectivity from the corresponding weight values (Fig. 3, bottom). This structure facilitates clean disentanglement of weights for voxels jointly selective for multiple concepts. The non-negativity constraint further sharpens interpretability by ensuring that positive weights indicate activation and negative weights indicate suppression. Together, these properties enable clear interpretation of conceptual selectivity across the cortex.

Both the Sparse and Dense Encoding Models can be interpreted under a Bayesian regression framework [35]. From this view, the Dense Model corresponds to the Sparse Model with a non-spherical Gaussian prior whose covariance captures correlations among concept directions in the dense space. A formal derivation is provided in the Appendix B.

## 3  Experiments

### 3.1  Voxelwise Modeling and Feature Construction

We evaluated the Sparse Concept Encoding Model on a publicly available fMRI dataset in which participants listened to naturalistic autobiographical audio stories [22]. Our analysis follows the voxelwise encoding model framework, emphasizing high-resolution, within-subject modeling in which model training and evaluation are performed on separate data from the same individual [15]. We present cortical maps for two representative participants and report group-level statistics across all seven participants.

To construct the dense feature space, we used pre-trained 300-dimensional GloVe word embeddings [41]. The corresponding sparse feature space was obtained using non-negative sparse dictionary learning, following the procedure of Zhang et al. [59]. Using their public implementation, we learned a 1,000-dimensional overcomplete dictionary of concept atoms from the GloVe vectors. For each atom, we identified the top-activating words from the story transcripts to aid interpretability. Figure 2

shows illustrative examples, and a broader list of 50 representative concept atoms is provided in Appendix C.

A key question is whether the dense feature space already provides sufficient semantic separation for voxel weights to be interpreted directly. If distinct concepts were represented along nearly orthogonal directions, sparse transformation would be unnecessary. To evaluate this, we examined the pairwise cosine similarities among the 1,000 learned concept atoms in the original 300-dimensional GloVe space. For reference, 1,000 random unit vectors in a 300-dimensional space are expected to be nearly orthogonal: their pairwise cosine similarities have mean $\mu = 0$, standard deviation $\sigma = 0.058$, and only $0.03\%$ of pairs exceed $0.2$ in magnitude. In contrast, the concept atoms exhibited substantial overlap, with mean pairwise similarity $\mu = 0.01$, standard deviation $\sigma = 0.075$, and $2.18\%$ of pairs exceeding $0.2$ (Fig. 4a). This structured correlation pattern indicates that the dense feature space represents concepts along moderately correlated directions, reaffirming the need for a sparse, axis-aligned representation.

### 3.2 Improving Embedding Geometry with Vector Norm Reparameterization

A known property of dense word embeddings is that high-frequency words tend to lie closer to the origin, likely because their vectors represent averages over more diverse contextual usages [44, 2]. We observe that sparse coding often captures many high-frequency words with different semantic meanings within the same dictionary atoms. To improve separability among high-frequency words, we developed a preprocessing technique called Vector Norm Reparameterization (VNR), which differentially rescales each word vector according to its norm, expanding the relative spread of vectors near the origin while compressing those among high-norm vectors. Quantitative comparisons with standard preprocessing methods (Appendix G) show that VNR improves WordNet purity, reconstruction error, and sparsity. Based on these results, we adopted VNR for all subsequent analyses.

### 3.3 Comparing Predictive Performance of Dense and Sparse Models

To compare the predictive performance and interpretability of the Dense Encoding Model and the Sparse Concept Encoding Model, we fit voxelwise encoding models using two types of semantic features: 300-dimensional dense embeddings for the Dense Model and 1,000-dimensional sparse embeddings for the Sparse Model. Both models were trained using regularized linear regression with cross-validation and evaluated on a held-out test story using the coefficient of determination ($R^2$) as the performance metric. Statistical significance was assessed at the voxel level using 9,999 block permutations with false discovery rate (FDR) correction at $p < 0.05$ (Appendix F). To visualize prediction performance, we plotted voxelwise $\sqrt{R^2}$ values onto cortical flatmaps [18]. The Sparse Model achieved strong prediction accuracy across temporal, parietal, and prefrontal cortices in both participants (Fig. 4b–c). To directly compare models, we plotted voxelwise $\sqrt{R^2}$ scores across all significantly predicted voxels (Fig. 4d–e). Across subjects, the mean difference in prediction performance between the transformed and original models was $-0.00047 \pm 0.00071$, with no statistically significant difference detected (paired $t(6) = -1.77$, $p = 0.13$). These results indicate that the Sparse Concept Encoding Model achieves comparable prediction accuracy to that of the Dense Encoding Model while substantially improving interpretability.

### 3.4 Concept-Level Cortical Maps from the Sparse Concept Encoding Model

Prior work has shown that abstract concepts are represented in the brain through widespread, spatially overlapping cortical patterns rather than anatomically localized regions [23, 22]. However, dense encoding models lack a principled way to disentangle the cortical representations of individual concepts. The Sparse Concept Encoding Model addresses this by representing each concept as a distinct axis in a sparse feature space. These axes, or concept atoms, correspond to interpretable semantic dimensions, which can be identified by inspecting the top-activating stimulus words (see Appendix C Table 1). In this and Section 3.6, we focus on the 20 atoms with the highest average cortical activation, as they provide the most robust signal for analyzing concept-level representations.

To assess whether the Sparse Concept Encoding Model yields consistent and interpretable cortical maps for individual semantic concepts, we examined two illustrative concept atoms, indices 456 and 115, drawn from the top 20 atoms (Fig. 5). These atoms are interpretable as representing the

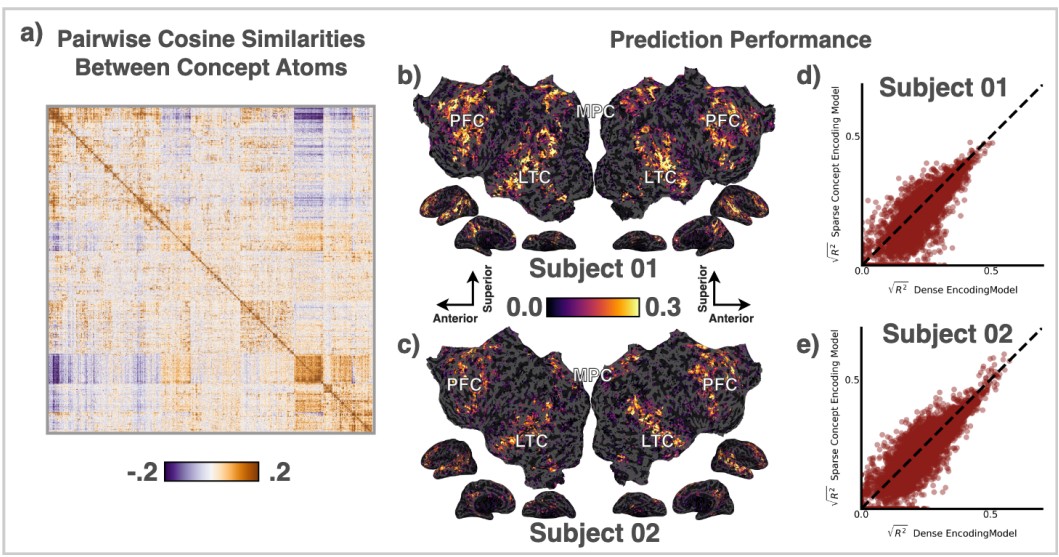

Figure 4: **The Sparse Concept Encoding Model preserves prediction accuracy while improving interpretability.** **(a)** Pairwise cosine similarities between all 1,000 concept atoms in the dense embedding space reveal structured correlations, with many atom pairs showing moderate cosine similarity. **(b-c)** Cortical maps of prediction accuracy ($\sqrt{R^2}$) for the Sparse Concept Encoding Model in Subject 01 and Subject 02, showing strong performance across lateral temporal (LTC), medial parietal (MPC), and prefrontal (PFC) regions. **(d–e)** Scatter plots comparing prediction accuracy between the Sparse and Dense Encoding Models across all significantly predicted voxels.

concepts of "body position" and "family", respectively. For each atom, we listed the most strongly activating stimulus words and, for two participants, visualized voxelwise regression weights along the corresponding concept-atom axis on cortical flatmaps. The resulting maps revealed bilateral, distributed patterns of selectivity that are consistent across individuals.

As a baseline, we compared these maps to those derived from the Dense Encoding Model. Since the dense model lacks a principled method for generating concept-level maps, prior work has relied on heuristic strategies. Common approaches include: (1) assigning each voxel to the concept whose embedding vector has the highest cosine similarity to its weight vector, which can underestimate mixed selectivity; and (2) projecting voxel weights onto individual semantic directions, which can overestimate selectivity due to correlations in dense space. In this work, we implemented a hybrid heuristic: labeling a voxel as selective for a concept if that concept ranked among its top-10 cosine similarity matches. Maps produced by the dense model were less spatially coherent, less bilaterally symmetric, and more variable across individuals (Appendix E).

To quantify inter-subject consistency across models, we projected each participant's cortical maps onto the common fsaverage surface and computed pairwise cosine similarities across all seven individuals. Averaged across the top 20 concept atoms, the Sparse Concept Encoding Model yielded substantially higher inter-subject consistency (mean $\pm$ SD : $0.26 \pm 0.04$) compared to the Dense Encoding Model ($0.09 \pm 0.04$). These results indicate that the sparse model produces more consistent semantic representations across subjects.

### 3.5 Case Study: Disentangling Time, Space, and Number Representations

A longstanding hypothesis in cognitive neuroscience posits that the brain encodes time, space, and number using a shared magnitude system [11, 53, 12]. While this theory is supported by robust behavioral and psychophysical evidence, neuroimaging results have been inconsistent [9, 48]. The Sparse Concept Encoding Model enables a novel test of this hypothesis by assigning time, space, and number independent axes in a shared semantic space, allowing direct comparison of their cortical representations. To evaluate this, we identified the concept atoms most associated with each domain based on their top-activating stimulus words (Fig. 6). To normalize for variation in

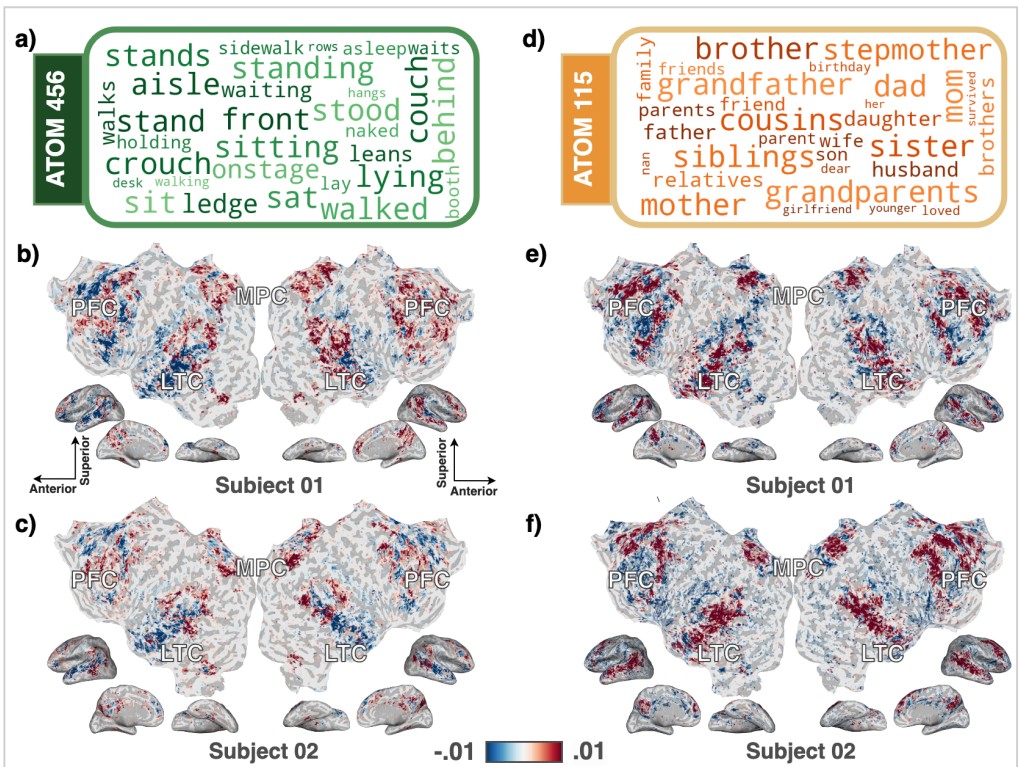

Figure 5: **The Sparse Concept Encoding Model yields consistent, interpretable cortical maps for individual semantic concepts.** (a, d) Two example concept atoms (indices 456 and 115) from the top 20 most strongly activated atoms, interpreted as representing the concepts of "body position" and "family," respectively. Each word cloud shows the top-activating stimulus words for the corresponding atom. (b–c, e–f) Voxelwise regression weights projected onto the selected concept-atom axis for Subject 01 and Subject 02. Red indicates positive selectivity and blue negative. Opacity is scaled by cross-validated prediction accuracy, and only voxels with significant prediction performance (jackknife-estimated $p < 0.05$) are shown. Both atoms exhibit distributed, bilaterally symmetric selectivity patterns that are consistent across individuals. These results show that the Sparse Concept Encoding Model successfully disentangles individual concept representations and produces coherent cortical maps.

activation magnitude across concept atoms, we selected the top 5,000 voxels with the highest positive regression weights for each cortical map. The resulting composite maps revealed localized, bilateral overlap among the three conceptual domains, indicated in white. These patterns were consistent across individuals. Notably, regions of three-way overlap included the bilateral intraparietal sulcus (IPS) and bilateral inferior frontal gyrus (IFG), areas previously implicated in shared magnitude processing [9, 48]. To quantify the extent of overlap, we calculated intersection-over-union (IoU) scores for each concept pair and their three-way intersection [49, 9, 48]. Averaged across seven participants, IoU scores were $0.34 \pm 0.05$ for time–space, $0.32 \pm 0.07$ for time–number, $0.32 \pm 0.07$ for space–number, and $0.18 \pm 0.05$ for voxels jointly selective for all three. These findings support the shared magnitude system hypothesis, offering fMRI evidence from naturalistic stimuli enabled by a novel and interpretable modeling framework.

### 3.6 Case Study: Comparing Cortical Representations of Different Concepts

Understanding how the brain encodes relationships between abstract concepts is a central goal in cognitive neuroscience. For example, are logical composition concepts such as "combine", "overlap", and "exclude" represented similarly to body position concepts like "lie," "sit," and "slouch"? To address this, we compared cortical tuning maps across the top 20 concept atoms. For each atom, we generated cortical maps as described in Section 3.4 and computed pairwise cosine similarities

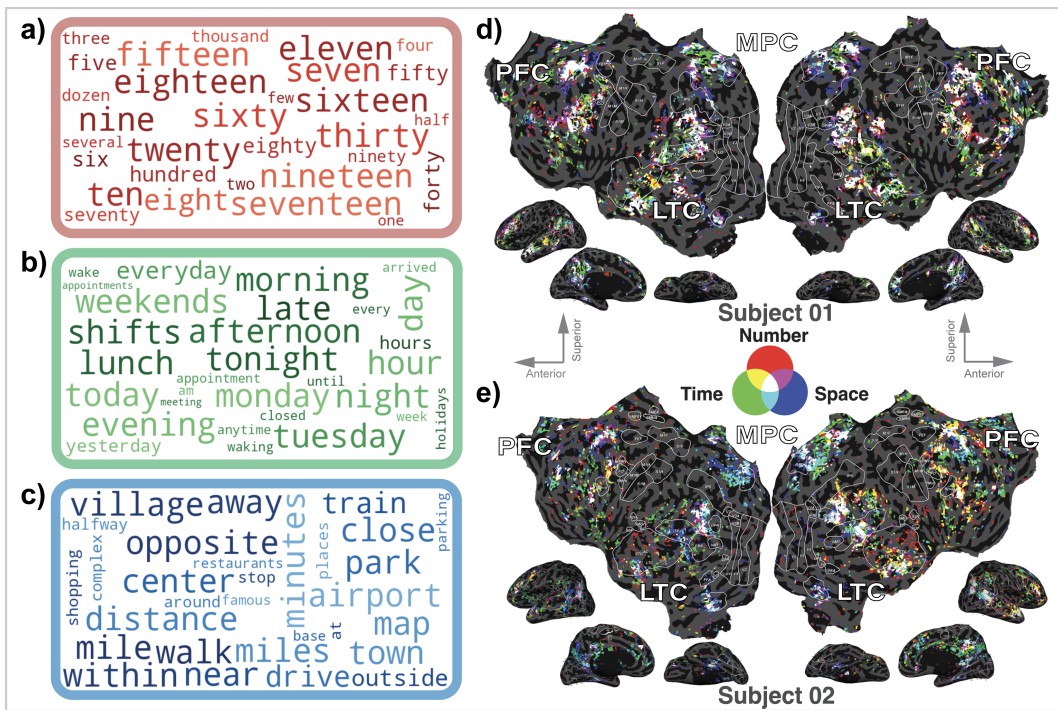

Figure 6: **Sparse Concept Encoding Model reveals overlapping cortical representations of time, space, and number.** **(a–c)** Word clouds showing the top 30 stimulus words that most strongly activate three selected concept atoms, corresponding to the abstract domains of number (red), time (green), and space (blue). **(d–e)** Composite cortical maps for Subject 01 and Subject 02 showing voxelwise selectivity to each concept atom. Each map displays the top 5,000 voxels with significant prediction performance and positive regression weights (jackknife-estimated $p < 0.05$) per atom. Red, green, and blue denote selectivity to number, time, and space, respectively; overlapping responses are shown in secondary colors, and white indicates joint selectivity to all three. Some inter-individual variation is present, as expected in within-subject functional maps derived from voxelwise models using naturalistic stimuli. These maps reveal localized, bilateral three-way overlap among time, space, and number representations, most notably in the intraparietal sulcus (IPS) and inferior frontal gyrus (IFG). This pattern is consistent with the hypothesis that these domains are represented by a shared magnitude system in the human brain.

between all maps, yielding a $20 \times 20$ concept-level similarity matrix. We applied hierarchical clustering to the similarity matrix from Subject 01 and visualized the reordered matrix alongside its dendrogram. Using the same atom ordering, we then plotted the similarity matrix for Subject 02. The clustering patterns were qualitatively consistent across subjects and quantitatively stable across all seven subjects: the mean off-diagonal Pearson correlation between similarity matrices across subjects was $0.70 \pm 0.08$. To ensure this structure was not inherited from the input embeddings, we computed pairwise Pearson correlations between sparse feature vectors across the 190 concept atom pairs. These correlations were uniformly low ($|r| = 0.00 \pm 0.10$; $85\% < 0.10$), confirming that the Sparse Concept Encoding Model effectively disentangles semantic directions. In contrast, the same concept directions in the original dense GloVe space showed positive moderate correlations ($0.16 \pm 0.14$; $87\% > 0$), suggesting dense embeddings obscure fine-grained distinctions between semantic directions. These findings suggest that the brain organizes abstract concepts according to a structured similarity geometry, which becomes observable once the feature representations are disentangled.

## 4 Discussion, Limitations, and Broader Impact

We introduced the Sparse Concept Encoding Model, a framework that addresses the interpretability challenges arising from superposition in dense feature spaces. Our experiments focused on fMRI

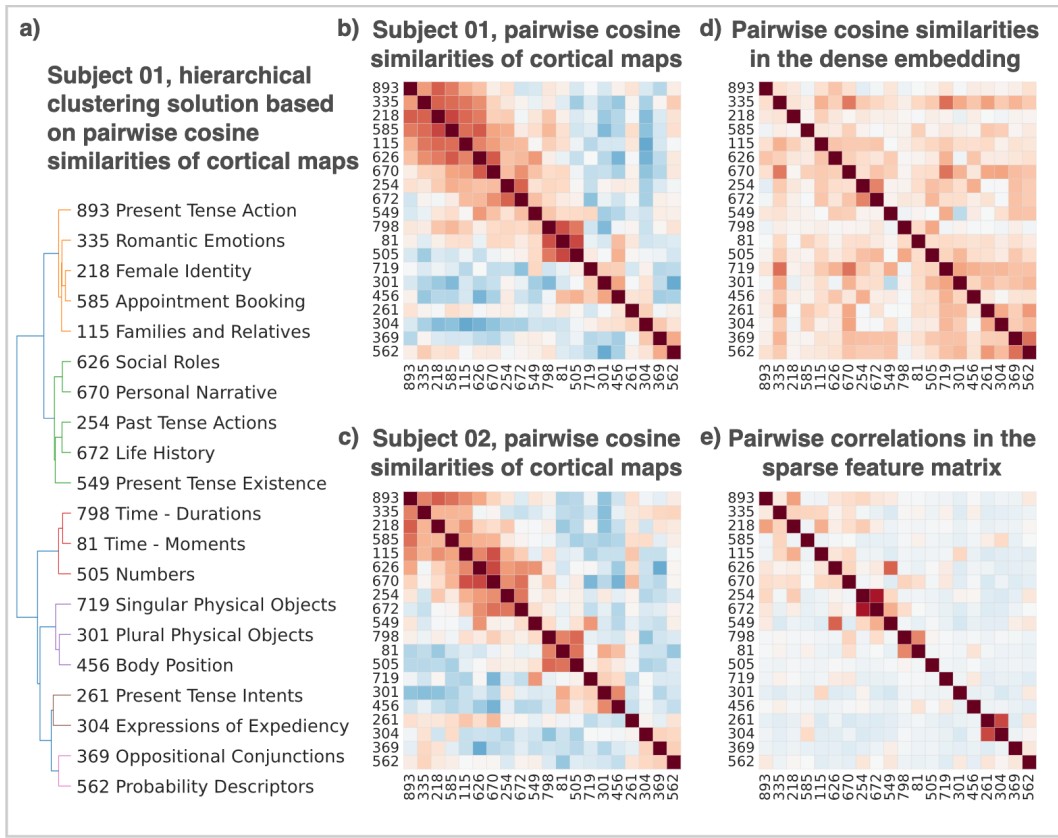

Figure 7: **Sparse Concept Encoding Model reveals consistent structure in concept-level cortical representations across subjects. (a)** Dendrogram from hierarchical clustering of Subject 01 based on pairwise cosine similarities between cortical tuning maps for the 20 most strongly represented concept atoms. Concept labels were heuristically assigned by summarizing the top-activating stimulus words for each atom. **(b, c)** Concept-level cortical representation similarity matrices for Subject 01 and Subject 02. Each matrix shows pairwise cosine similarity between cortical maps for the 20 concept atoms, reordered using the clustering from Subject 01. The structure is qualitatively consistent across subjects, indicating shared cortical representational structure. **(d)** Cosine similarity matrix between the same 20 atoms in the original dense GloVe embedding space, showing structured correlations that would confound interpretation under the Dense Encoding Model. **(e)** Pearson correlation matrix of the sparse feature vectors used to train the Sparse Concept Encoding Model. Off-diagonal values are uniformly low, confirming that the sparse transformation yields disentangled semantic dimensions.

data and static word embeddings, but the framework is broadly applicable to domains requiring interpretable mappings from entangled features. One current limitation is that the model does not incorporate linguistic context, as it relies on non-contextual word embeddings. Extending it to contextualized representations from large language models is a promising direction for future work [24].

Although concept atoms improve interpretability, they may inherit structural biases from the original embeddings, potentially amplifying harmful stereotypes. Moreover, while this method could enhance brain–computer interface technologies by enabling more accurate semantic decoding, it raises ethical concerns around mental privacy and the potential misuse of neural decoding tools [19].

## Acknowledgments and Disclosure of Funding

We thank Bruno Olshausen and Yubei Chen for helpful discussions. This work was supported in part by the National Eye Institute (NEI) of the National Institutes of Health under award R01 EY031455.

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

# A  Experimental Data and Encoding Model Pipeline

## A.1  Data Availability

The raw fMRI dataset used in this study is publicly available via the Gallant Lab GIN repository.[1] Preprocessed fMRI data and aligned stimulus features are also available on OSF.[2]

All data used in this study—excluding anatomical MRI scans—have been made publicly accessible. Anatomical images were withheld to protect participant privacy. To ensure full reproducibility, we provide transformation matrices that map volumetric data to individual cortical flatmaps, enabling accurate surface-based visualization of the results.

## A.2  Experimental Procedure

The dataset used in this study was originally collected for a previously published experiment [22]. All procedures related to participant recruitment, scanning parameters, and data acquisition were described in that work. Briefly, each participant completed eleven fMRI scans while listening to narrative audio stories from *The Moth Radio Hour*, in which individuals recount autobiographical experiences before a live audience. Each scan featured a single story lasting 10–15 minutes, with 10 seconds of silence preceding and following the audio.

## A.3  Dataset Structure and Test/Train Split

The full dataset comprised eleven stories per participant. Ten stories were used for training the encoding models, yielding approximately 125 minutes of data per subject. One remaining story was held out for testing and presented twice to each participant. To improve the signal-to-noise ratio in the test data, BOLD responses across the two repetitions were averaged. Final model evaluations were conducted on this averaged 10-minute test set.

## A.4  fMRI Preprocessing

Functional MRI data were preprocessed using the same pipeline as described in a previously published study [31]. The resulting fMRI response matrices consisted of approximately 60,000–80,000 voxels per subject. All analyses were conducted in each subject's native voxel space, with no spatial smoothing applied.

## A.5  Feature Construction

**Low-Level Acoustic Features.**  To control for correlations between semantic content and acoustic structure, we included 41 nuisance regressors following [13]:

- 39 phoneme-rate features reflecting the frequency of each English phoneme over time
- A single feature representing total phoneme rate
- A single feature representing word rate

**Semantic Features.**  Each word in the stimulus transcript was transformed into two parallel semantic representations:

- 300-dimensional dense GloVe embeddings [41]
- 1,000-dimensional sparse concept embeddings derived via dictionary learning from GloVe

All features were resampled to match the fMRI acquisition rate using a 3-lobe Lanczos low-pass filter with a cutoff of 0.249 Hz (the Nyquist frequency). This resulted in training feature matrices of size $3{,}717 \times 300$ for the Dense Encoding Model and $3{,}717 \times 1{,}000$ for the Sparse Concept Encoding Model.

---

[1]`https://gin.g-node.org/gallantlab/story_listening`
[2]`https://osf.io/5dvpy/?view_only=e0801550057746b5a1713a96e05a11ff`

To account for hemodynamic lag, a finite impulse response (FIR) model was applied to the design matrix. Each feature time series was delayed by 1, 2, 3, and 4 TRs (corresponding to 2, 4, 6, and 8 seconds) and concatenated, allowing the model to estimate a linear temporal kernel per feature [15].

## A.6 Feature Normalization

All stimulus features were demeaned over time within each scan run, except for the 300-dimensional GloVe embeddings. These embeddings were left unchanged to maintain a consistent origin across runs. Because cosine similarity depends on vector direction relative to a fixed origin, shifting the origin across runs would undermine interpretability. In contrast, sparse concept features and low-level acoustic features were demeaned. For these features, mean-centering does not affect the interpretation of voxel weights.

## A.7 Model Estimation

Voxelwise encoding models were estimated using banded ridge regression [14], which applies separate regularization to the semantic and low-level feature subspaces. For each voxel, two regularization hyperparameters ($\lambda_{\text{semantic}}$ and $\lambda_{\text{low-level}}$) were optimized using leave-one-run-out cross-validation on the training set.

The procedure was as follows:

- The 10 training runs were split into 9 training and 1 validation run, iterated over all folds.
- For each fold, 20 logarithmically spaced values (from $10^1$ to $10^{20}$) were tested for each hyperparameter.
- Prediction accuracy was computed on the held-out run.
- The hyperparameter pair yielding the highest average prediction accuracy across folds was selected separately for each voxel.

Model fitting was implemented using the `Himalaya` Python library [14][3], which enables efficient banded ridge regression on both CPUs and GPUs. Training required approximately 30–60 minutes per subject on a single NVIDIA RTX A6000 GPU.

## A.8 Model Evaluation

Final models were evaluated on the held-out test story using the average blood-oxygen-level-dependent (BOLD) responses from two repeated presentations. Prediction accuracy was computed as the coefficient of determination ($R^2$) between predicted and observed fMRI responses, independently for each voxel.

## A.9 Postprocessing of Semantic Weights

Model weights were first averaged across the five FIR delays to yield a single value per feature per voxel. To remove differences in overall scale introduced by voxel-specific ridge regularization, each semantic weight vector was normalized to unit length. Vectors were then scaled by the voxel's cross-validated prediction accuracy, allowing downstream analyses to reflect both tuning direction and model reliability. Negative accuracies were clipped to zero before scaling.

## A.10 Software Implementation

All analyses were implemented in Python using a custom framework called `sparseconcept`, which is openly available on GitHub.[4] The analysis pipeline relies on standard scientific Python libraries including `numpy`, `scipy`, `matplotlib`, `scikit-learn`, `statsmodels` and `pycortex`, and uses the `himalaya` package with a PyTorch backend for efficient voxelwise model fitting.

---

[3]`https://github.com/gallantlab/himalaya`
[4]`https://github.com/alicialitrtwe/sparseconcept`

# B    Non-identifiability of the Dense Encoding Model and Bayesian Interpretation of Dense vs. Sparse Encoding Models

## B.1    Theorem (Non-identifiability in Dense Embedding Spaces)

Let $\{\mathbf{f}_i\}_{i=1}^k \subset \mathbb{R}^d$ be unit-norm concept atoms, and define the Gram matrix:

$$G = [G_{ij}]_{i,j=1}^k, \quad G_{ij} = \langle \mathbf{f}_i, \mathbf{f}_j \rangle.$$

Suppose a voxel's weight vector is tuned to a linear combination of these atoms:

$$\mathbf{w} = \sum_{i=1}^k \alpha_i \, \mathbf{f}_i, \qquad \boldsymbol{\alpha} = (\alpha_1, \ldots, \alpha_k)^\top.$$

Taking inner products with each $\mathbf{f}_j$ yields the observed projections:

$$\begin{pmatrix} \langle \mathbf{w}, \mathbf{f}_1 \rangle \\ \vdots \\ \langle \mathbf{w}, \mathbf{f}_k \rangle \end{pmatrix} = G\boldsymbol{\alpha}.$$

If $G$ is invertible, then $\boldsymbol{\alpha}$ can be uniquely recovered. However, if $G$ is singular, the system is non-identifiable: multiple combinations of concept atoms can yield the same weight vector. In the ideal case where all $\mathbf{f}_j$ are orthogonal, $G = I$ and $\boldsymbol{\alpha}$ can be directly recovered.

## B.2    Bayesian Interpretation of Sparse and Dense Encoding Models

Both the Sparse Concept Encoding Model and the Dense Encoding Model admit a Bayesian linear regression interpretation [35]. This view makes explicit the prior assumptions each model imposes on voxel tuning over semantic features.

**Sparse Concept Encoding Model.**    Each stimulus is represented by a sparse, non-negative activation over $m$ learned concept atoms, yielding a design matrix $Z \in \mathbb{R}^{T \times m}$. Voxel responses are fit with ridge regression

$$\hat{\boldsymbol{\beta}} = \arg \min_{\boldsymbol{\beta} \in \mathbb{R}^m} \|\mathbf{y} - Z\boldsymbol{\beta}\|_2^2 + \lambda \|\boldsymbol{\beta}\|_2^2,$$

which corresponds to MAP inference under an isotropic Gaussian prior

$$\boldsymbol{\beta} \sim \mathcal{N}(0, \lambda^{-1} I).$$

This prior assumes independent contributions of each concept atom with equal variance.

**Dense Encoding Model.**    Let the dense feature matrix be factorized as $X_{\text{dense}} = Z\Phi$, where $\Phi \in \mathbb{R}^{m \times d}$ contains the concept atoms as row vectors in the dense embedding space, and fit ridge regression in dense space:

$$\hat{\mathbf{w}} = \arg \min_{\mathbf{w} \in \mathbb{R}^d} \|\mathbf{y} - Z\Phi\mathbf{w}\|_2^2 + \lambda \|\mathbf{w}\|_2^2.$$

Define the implied sparse-space weights $\boldsymbol{\beta} := \Phi\mathbf{w} \in \mathbb{R}^m$. As shown below, dense ridge is equivalent to a sparse-space regression with a structured (generally degenerate) Gaussian prior

$$\boldsymbol{\beta} \sim \mathcal{N}(0, \lambda^{-1} \Phi\Phi^\top).$$

When $\Phi\Phi^\top$ is singular, this is a degenerate Gaussian supported on $\mathcal{C}(\Phi)$; its MAP penalty is $\boldsymbol{\beta}^\top (\Phi\Phi^\top)^+ \boldsymbol{\beta}$. Intuitively, atoms with similar directions (large dot product) will be biased to have coupled weights under the prior. This coupling can improve prediction when related concepts truly co-activate, but it complicates interpretation: voxel weights may reflect embedding correlations rather than true semantic selectivity.

### B.2.1 Proof

Start from dense ridge:
$$\hat{\mathbf{w}} = \arg \min_{\mathbf{w} \in \mathbb{R}^d} \|\mathbf{y} - Z\Phi\mathbf{w}\|_2^2 + \lambda\|\mathbf{w}\|_2^2.$$

Let $\boldsymbol{\beta} := \Phi\mathbf{w}$. Then $\|\mathbf{y} - Z\Phi\mathbf{w}\|_2^2 = \|\mathbf{y} - Z\boldsymbol{\beta}\|_2^2$, and feasible $\boldsymbol{\beta}$ satisfy $\boldsymbol{\beta} \in \mathcal{C}(\Phi)$.

**Case A:** $\mathrm{rank}(\Phi) = d$ (injective). For each $\boldsymbol{\beta} \in \mathcal{C}(\Phi)$ there is a unique $\mathbf{w}$ such that $\Phi\mathbf{w} = \boldsymbol{\beta}$, given by
$$\mathbf{w} = (\Phi^\top\Phi)^{-1}\Phi^\top\boldsymbol{\beta}.$$

Substituting into the objective yields
$$\hat{\boldsymbol{\beta}} = \arg \min_{\boldsymbol{\beta} \in \mathcal{C}(\Phi)} \|\mathbf{y} - Z\boldsymbol{\beta}\|_2^2 + \lambda\boldsymbol{\beta}^\top(\Phi\Phi^\top)^+\boldsymbol{\beta}.$$

If $m = d$ and $\Phi$ is invertible, then $\mathcal{C}(\Phi) = \mathbb{R}^m$ and $(\Phi\Phi^\top)^+ = (\Phi\Phi^\top)^{-1}$.

**Case B:** $\mathrm{rank}(\Phi) < d$ (non-injective). For fixed $\boldsymbol{\beta} \in \mathcal{C}(\Phi)$, the constraint $\Phi\mathbf{w} = \boldsymbol{\beta}$ admits infinitely many solutions. Moreover, all $\mathbf{w}$ satisfying $\Phi\mathbf{w} = \boldsymbol{\beta}$ yield the same prediction $Z\Phi\mathbf{w} = Z\boldsymbol{\beta}$, so the data-fit term is constant over this solution set. Dense ridge therefore selects the solution with minimal norm,
$$\mathbf{w}_{\min} = \arg \min_{\mathbf{w}:\,\Phi\mathbf{w}=\boldsymbol{\beta}} \|\mathbf{w}\|_2^2,$$

which is given by the Moore–Penrose pseudoinverse,
$$\mathbf{w}_{\min} = \Phi^\dagger\boldsymbol{\beta}, \qquad \Phi^\dagger = (\Phi^\top\Phi)^+\Phi^\top.$$

Substituting $\mathbf{w}_{\min}$ into the regularizer gives
$$\min_{\mathbf{w}:\,\Phi\mathbf{w}=\boldsymbol{\beta}} \|\mathbf{w}\|_2^2 = \|\Phi^\dagger\boldsymbol{\beta}\|_2^2 = \boldsymbol{\beta}^\top(\Phi\Phi^\top)^+\boldsymbol{\beta},$$

and thus recovers the same sparse-space objective as in Case A:
$$\hat{\boldsymbol{\beta}} = \arg \min_{\boldsymbol{\beta} \in \mathcal{C}(\Phi)} \|\mathbf{y} - Z\boldsymbol{\beta}\|_2^2 + \lambda\boldsymbol{\beta}^\top(\Phi\Phi^\top)^+\boldsymbol{\beta}.$$

## C  Top 50 Sparse Concept Atoms and Associated Words

We identified the 50 concept atoms most strongly expressed in the Sparse Concept Encoding Model, based on their average activation across voxels. During analysis, we found that a small number of atoms exhibited high activation correlations with many others in the sparse feature matrix. This redundancy complicates interpretation by making it difficult to isolate distinct conceptual contributions to neural responses. To address this, we excluded highly redundant atoms from all subsequent analyses. Specifically, we computed the pairwise Pearson correlation matrix across all 1,000 sparse feature dimensions and calculated the average absolute correlation of each atom with all others. Atoms whose average correlation exceeded 70% of the maximum observed value were removed. This thresholding procedure yielded a decorrelated subset of concept atoms better suited for interpreting voxelwise model weights. From the remaining atoms, we computed the mean absolute activation across voxels using the postprocessed model weights. The 50 atoms with the highest average activations were selected, as they were most strongly engaged across the brain. To interpret the semantic content of each atom, we identified the most strongly associated words from the training stories. Only words with an activation value greater than 0.2 for a given atom were retained. For each selected atom, the 20 highest-ranked words (sorted by activation strength) are listed in Table 1, along with the atom's original index from the full 1,000-dimensional dictionary.

Table 1: Fifty concept atoms with the highest average cortical activation, listed with their top associated words.

| Index | Top Associated Words |
|---|---|
| 505 | eleven, sixteen, nineteen, fifteen, seventeen, twenty, ten, eighteen, thirty, sixty |
| 562 | usually, often, also, normally, may, actually, always, might, definitely, sometimes |
| 254 | jumped, ran, came, pulled, blew, proceeded, took, stood, fell, stopped |
| 115 | grandparents, sister, dad, cousins, brother, siblings, mom, grandfather, mother, stepmother |
| 261 | learn, want, able, take, join, to, need, begin, wish, go |
| 585 | call, telephone, appointment, phone, us, anytime, ask, queries, answering, number |
| 670 | business, personal, home, city, family, school, music, game, woman, life |
| 369 | although, though, however, despite, but, yet, anyway, surprisingly, somewhat, far |
| 549 | is, becomes, an, itself, a, exists, another, comes, represents, was |
| 301 | cars, dolls, sticks, boxes, pillows, shirts, cans, beds, are, cards |
| 218 | herself, queen, goddess, her, waitress, she, sister, mother, woman, whore |
| 672 | died, since, graduated, moved, dated, began, until, lived, existed, was |
| 304 | easy, easier, quick, simple, instantly, allows, handy, lets, solution, fast |
| 456 | standing, sitting, stood, sat, sit, crouch, aisle, stands, lying, behind |
| 798 | months, hours, days, weeks, ago, years, minutes, hour, month, year |
| 719 | cow, bird, hat, cat, tiny, bag, ball, pink, baby, tree |
| 893 | pulls, sees, leans, asks, grabs, takes, waits, thinks, goes, gets |
| 81 | morning, afternoon, evening, weekends, monday, tuesday, lunch, night, hour, day |
| 626 | guy, sailor, boy, woman, waitress, man, doctor, stranger, mister, kid |
| 335 | cry, heart, eyes, soul, breath, lover, alive, dream, cruel, smile |
| 172 | acknowledged, insisted, suggested, explained, said, admitted, saying, recounted, claimed, told |
| 707 | thanks, hello, sorry, hey, bye, hi, dear, thank, yeah, ya |
| 530 | am, wrote, by, at, on, subject, anonymous, views, writes, abuse |
| 184 | discuss, topic, explored, similarities, about, examining, issues, examine, subject, dealing |
| 148 | toward, towards, forward, fro, slowly, direction, moving, into, back, headed |
| 486 | steve, dave, andy, tim, rob, kevin, matt, bob, todd, nick |
| 733 | from, this, notes, ago |
| 244 | insanely, remarkably, incredibly, terribly, extremely, perennially, oddly, horribly, surprisingly, quite |
| 494 | gon, outta, ya, gotta, lotta, em, wanna, gonna, fo, shit |
| 17 | page, here, visit, details, check, listed, list, information, above, see |
| 997 | near, close, distance, airport, center, miles, opposite, minutes, village, mile |
| 385 | um, o, em, eh, e, legal, ate, nova, do |
| 750 | many, other, several, common, similar, different, such, particularly, most, certain |
| 321 | awesome, great, amazing, lovely, nice, beautiful, gorgeous, brilliant, definitely, neat |
| 384 | talking, doing, enjoying, pushing, jumping, stopping, taking, chasing, throwing, keeping |
| 766 | everywhere, anywhere, happening, in, within, exist, occurs, occurred, exists, somewhere |
| 782 | brother, himself, father, son, his, grandfather, marquis, him, minister, afterwards |
| 640 | families, residents, people, adults, students, women, children, relatives, men, fellow |
| 890 | director, vice, professor, consultant, assistant, manager, says, said, joined, counselor |
| 730 | teachers, school, teacher, kindergarten, classroom, grades, grade, education, students, gifted |
| 610 | drank, counseled, listened, cared, worshipped, hated, travelled, liked, ate, lived |
| 867 | hands, finger, arms, arm, cheek, shoulders, thumb, rubbing, squeezes, ears |
| 8 | caused, cause, due, occurs, result, serious, suffered, failure, loss, problems |
| 119 | shorts, shirts, jeans, shirt, pants, jacket, cropped, vest, lauren, casual |
| 264 | bedroom, upstairs, room, downstairs, bathroom, beds, lobby, floor, closet, living |
| 886 | stabbed, stabbing, victim, fled, suspicious, blaze, killed, raped, man, missing |
| 970 | our, proud, committed, recognize, mission, ourselves, deeply, understands, tradition, honor |
| 853 | thin, tricky, slightly, tight, somewhat, bit, awkward, too, notoriously, low |
| 700 | alabama, virginia, arizona, florida, california, georgia, texas, vermont, washington, wyoming |
| 842 | event, held, attended, events, conference, booth, invited, host, weekend, fair |

# D   Cortical Representations of Time, Number, and Space

## D.1   Semantic Category Selection

To identify concept atoms associated with the semantic domains of time, number, and space, we began by defining representative seed word lists for each category. Time-related words included *second*, *hour*, *day*, *week*, *month*, *year*, *today*, and *ago*; number-related terms included *one*, *two*, *three*,

*ten*, *fifteen*, *hundred*, and *thousand*; and spatial expressions included *near*, *far*, *distance*, *mile*, *outside*, *map*, and *direction*.

For each category, we ranked all 1,000 concept atoms by their average activation across the corresponding seed words and manually examined the top five. We retained atoms whose top associated words were semantically aligned with the target category, using inclusive criteria to ensure coverage of relevant conceptual variations. Table 2 summarizes the selected concept atoms along with their highest-activation words.

Table 2: Concept atoms with strongest activation for semantic categories of time, number, and space.

| Factor Index | Category | Top Associated Words |
|---|---|---|
| 81 | Time | morning, afternoon, evening, weekends, monday, tuesday, lunch, night, hour, day, tonight, today, late, shifts, everyday, yesterday, hours, appointment, am, closed |
| 798 | Time | months, hours, days, weeks, ago, years, minutes, hour, month, year, week, day, sec, seconds, spent, intervening, semester, times, last, weekends |
| 880 | Time | seventies, sixties, twenties, late, earliest, grunge, subsequent, years, tenure, fame, nineteen, persisted, prior, later, existed, acclaim |
| 394 | Number | double, single, two, pair, simultaneous, four, three |
| 505 | Number | eleven, sixteen, nineteen, fifteen, seventeen, twenty, ten, eighteen, thirty, sixty, seven, nine, eight, eighty, six, fifty, five, forty, hundred, seventy |
| 602 | Number | digs, career, nine, tied, seven, sixth, eight, points, fifth, record, freshman, impressive, five, six, mark, finishes, shy, leads, four, straight |
| 353 | Space | threshold, average, calculated, absolute, scale, higher, slope, humidity, altitude, amount, low, normal, mean, distance, lower, level, speed, above, cumulative, readings |
| 997 | Space | near, close, distance, airport, center, miles, opposite, minutes, village, mile, town, park, map, away, within, walk, train, drive, outside, halfway |

These concept atoms capture distinct subtypes of meaning within each semantic domain. For example, factor 81 emphasizes time points typically used in "*at*" contexts (e.g., *at night*, *at 9 a.m.*); factor 798 encodes temporal durations often found in "*for*" constructions (e.g., *for days*, *for weeks*); and factor 880 reflects temporal eras or historical references (e.g., *seventies*, *grunge*, *persisted*).

For the main analysis in Section 3.5, we selected one representative concept atom from each semantic category—factor 81 (time), factor 505 (number), and factor 997 (space). While our primary results focus on these exemplars, preliminary inspection of other atoms within the same semantic domains suggests they evoke qualitatively similar cortical activation patterns. To assess the similarity of these representations more systematically, we computed cosine similarities between cortical weight maps for all pairwise combinations within three semantic groups: Time (81, 798, 880), Number (394, 505), and Space (353, 997). Number (394–505) and Space (353–997) pairs exhibited consistently high similarity ($0.502 \pm 0.063$ and $0.495 \pm 0.088$, respectively), whereas Time pairs showed moderate consistency (e.g., 81–798: $0.431 \pm 0.138$; 798–880: $0.457 \pm 0.098$). These within-group similarities were substantially higher than the background distribution of across-group factor pairs, which averaged just $0.048 \pm 0.013$. These findings provide quantitative support for the functional clustering of concept atoms within semantic domains and highlight the potential for future research to explore the structure and organization of these representational spaces more deeply.

## E Heuristic Concept-Level Cortical Maps from the Dense Encoding Model

This appendix presents heuristic concept-specific cortical maps generated from the Dense Encoding Model, used for qualitative comparison against the Sparse Concept Encoding Model, as discussed in Sections 3.5 and 3.6. Because the Dense Encoding Model produces voxel weight vectors in a

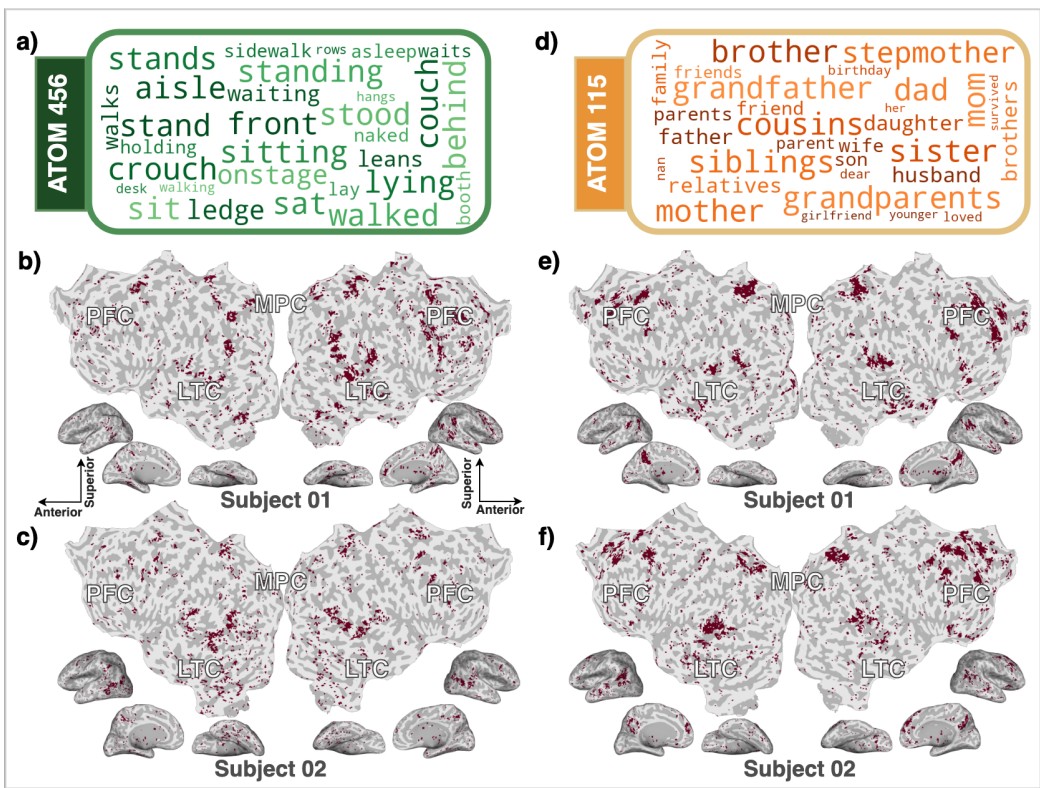

Figure 8: Heuristic concept-level cortical maps from the Dense Encoding Model for concept atoms 456 (a–c) and 115 (d–f), shown for Subject 01 and Subject 02. Panels (a, d) display word clouds for the top words associated with each atom. Panels (b–c, e–f) show voxelwise cortical maps, thresholded using the heuristic method described in Appendix E. Compared to the corresponding maps in Section 3.4 (Fig. 5), these maps are less spatially coherent and more variable across subjects, highlighting the interpretability gains achieved by modeling directly in sparse concept space.

continuous, entangled semantic space, it does not assign explicit concept axes. To approximate the cortical representation of a specific concept atom, we applied the following heuristic procedure:

**Select target concept atom.** We began with a target concept atom and retrieved its top 30 activating words using the learned sparse factor embeddings.

**Project dense embeddings onto voxel weights.** We loaded voxel weights from the Dense Encoding Model. The mean dense embedding of the selected concept atom's top words was computed and projected onto each voxel's weight vector, yielding an estimate of that voxel's tuning strength to the concept.

**Select significant voxels.** We retained voxels that satisfied both of the following: (a) positive mean cross-validated prediction scores, and (b) a positive projection for the selected concept. This step produced a subset of candidate voxels potentially selective for the concept.

**Label voxels by semantic similarity.** For each candidate voxel, we computed cosine similarity between its weight vector and all 1,000 sparse concept atoms. We then identified the top 10 most similar atoms for each voxel.

**Mask concept-relevant voxels.** Voxels were marked as selective for the concept if the original target atom appeared among their top-10 matches.

This procedure was applied independently for each concept examined in Sections 3.5 and 3.6. While these heuristic maps provide a rough approximation of concept tuning in the dense model, they are

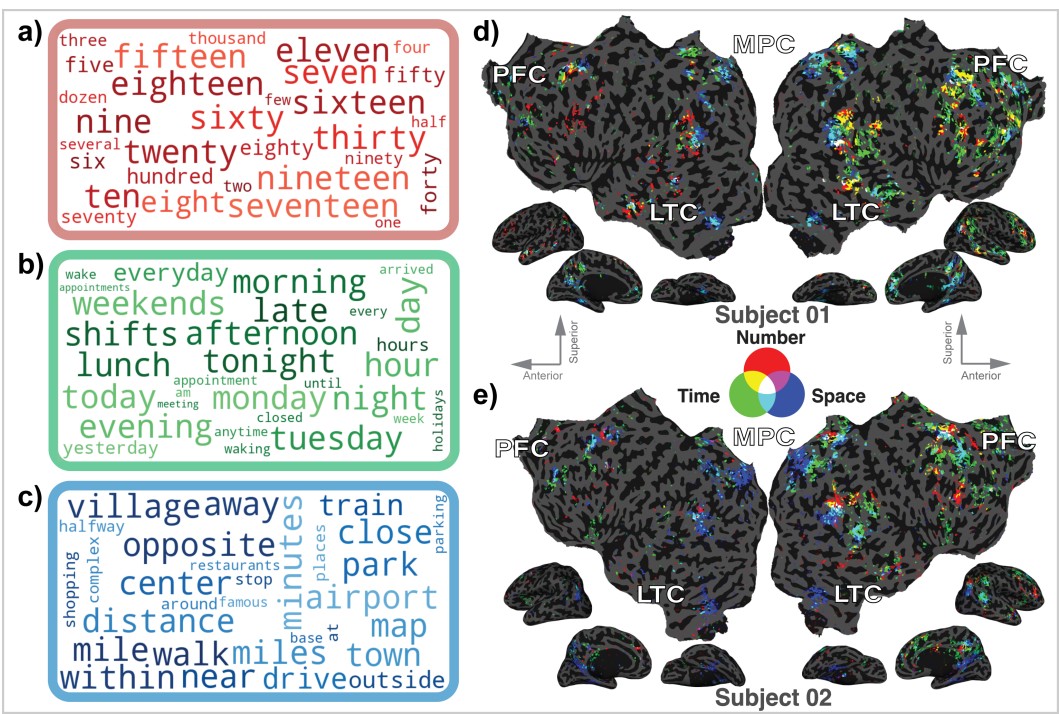

Figure 9: Heuristic composite cortical maps from the Dense Encoding Model for three concept atoms representing number (red), time (green), and space (blue). Word clouds (a–c) show the top stimulus words most strongly associated with each concept. Panels (d–e) show voxelwise cortical maps for Subject 01 and Subject 02. Colors indicate voxels selectively tuned to each concept (red, green, blue) or their overlaps (white indicating joint selectivity). This figure parallels the analysis in Figure 6 of the main paper (Section 3.5), but reveals substantially less spatial overlap between concepts, highlighting the limitations of the heuristic method.

less spatially coherent, less symmetric across hemispheres, and more variable across individuals than maps produced by the Sparse Concept Encoding Model. These differences, visualized in Figures 8 and 9, underscore the interpretability gains afforded by modeling directly in sparse concept space.

# F   Statistical Significance Maps from Permutation Testing

This appendix presents binary voxelwise significance maps obtained using non-parametric permutation testing to assess prediction accuracy on the test set. To construct a voxelwise null distribution, we performed non-parametric permutation testing. Specifically, the joint predictions of the banded ridge regression model, defined as the sum of predictions from the lexical-semantic and low-level acoustic feature spaces, were randomly permuted over time within each voxel. To account for temporal autocorrelation in the BOLD signal, we used a block permutation strategy following [13], permuting non-overlapping blocks of 10 consecutive time points (equivalent to 20 seconds of data). For each permutation, prediction accuracy was recomputed, and this process was repeated 9,999 times to generate a null distribution of accuracy values for each voxel. An empirical $p$-value was then computed per voxel by counting the number of permutations in which the permuted prediction accuracy exceeded the observed accuracy from the original (unpermuted) test data. To avoid zero-valued $p$-values, we added one to both the numerator and the denominator, following the method of [42]. To correct for multiple comparisons across voxels, the empirical $p$-values were adjusted using the Benjamini–Hochberg false discovery rate (FDR) procedure [6]. FDR correction was applied using the `multipletests` function from the `statsmodels` Python package with `method='fdr_bh'`. Finally, a binary mask of statistically significant voxels was generated by thresholding the FDR-corrected $p$-value map at $p < 0.05$.

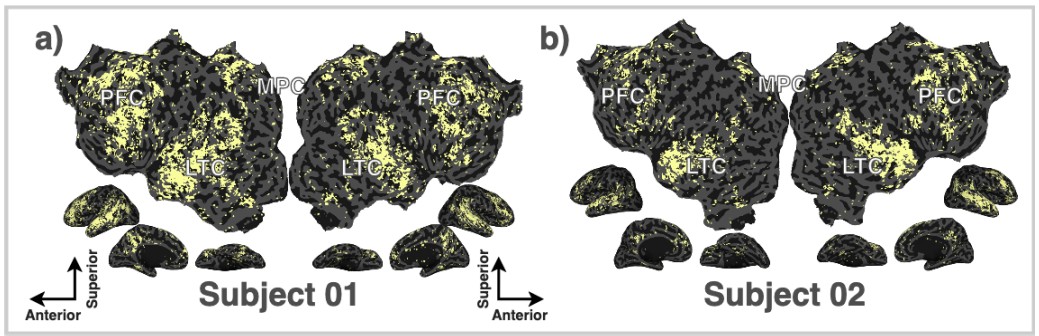

Figure 10: Binary statistical significance maps of prediction accuracy for the Sparse Concept Encoding Model in Subject 01 and Subject 02, highlighting significant prediction performance in temporal, parietal, and prefrontal regions.

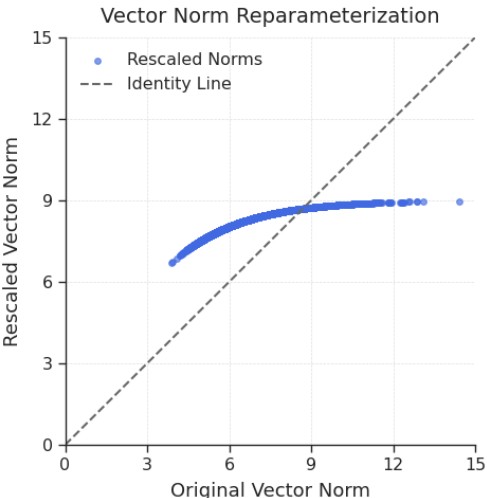

Figure 11: Effect of VNR on embedding norms. Each point represents a word vector before and after applying VNR. The dashed line indicates the identity (no change). VNR is a preprocessing method that enhances the interpretability of learned concept atoms by expanding the relative spread of vectors near the origin while compressing those among high-norm vectors.

## G   Evaluation of Embedding Preprocessing Methods

While most learned concept atoms captured coherent semantic themes, we observed that a subset activated broadly across many high-frequency words used in disparate contexts. We hypothesize that this ambiguity stems from a known property of word embeddings: high-frequency words tend to have smaller norms and cluster near the origin in Euclidean space, reducing their separability. To address this, we developed a custom preprocessing method called **Vector Norm Reparameterization (VNR)**. VNR improves the geometric structure of the embedding space by nonlinearly rescaling each word vector's norm while preserving its direction (Fig. 11). For a word vector $\mathbf{v}$, we define the scale factor as $s = 9 \cdot \tanh\left(\frac{2\|\mathbf{v}\|}{9} + \frac{1}{10}\right)$, and the reparameterized vector becomes $\tilde{\mathbf{v}} = \frac{\mathbf{v}}{\|\mathbf{v}\|} \cdot s$.

This transformation compresses large norms more than small ones, effectively expanding semantically dense regions and reducing the influence of rare words.

To evaluate the effectiveness of VNR, we compared it against three alternative preprocessing strategies applied prior to dictionary learning:

1. Unmodified GloVe vectors
2. L2 normalization to the unit hypersphere
3. ZCA whitening

Each method was evaluated using the following metrics:

**Reconstruction Error.** We computed the relative Frobenius reconstruction error $\|X_{\text{dense}} - Z\Phi\|_F/\|X_{\text{dense}}\|_F$ where $X_{\text{dense}} \in \mathbb{R}^{T \times d}$ contains the original dense word-embedding features, $Z \in \mathbb{R}^{T \times m}$ is the learned sparse, non-negative concept-atom activation matrix, and $\Phi \in \mathbb{R}^{m \times d}$ contains the corresponding concept atoms (rows of $\Phi$). This metric measures how well the sparse concept representation reconstructs the dense embedding space.

**Sparsity.** For each word, we calculated the fraction of zero entries in its 1,000-dimensional sparse code, then averaged this value across the stimuli vocabulary. Higher values indicate sparser representations.

**WordNet Purity.** To assess semantic coherence, we identified the top 50 highest-activation words for each concept atom and computed the mean pairwise WordNet similarity among them, using both path similarity and Wu–Palmer similarity [33, 40]. Semantically meaningful atoms are expected to group together words from similar lexical fields, as defined in WordNet's ontology.

**Results.** As shown in Table 3, VNR achieved the lowest reconstruction error while maintaining high sparsity and the strongest semantic coherence across both WordNet-based metrics. Notably, L2 normalization resulted in extremely poor reconstruction performance, likely due to its constraint that all word vectors and dictionary atoms lie on the unit hypersphere, limiting the model's ability to encode useful norm variation. Based on these results, we adopted VNR as the default preprocessing method in all subsequent experiments.

Table 3: Comparison of preprocessing strategies for word embeddings prior to sparse coding.

| Embedding Version | Reconstruction Error | Sparsity | WN Purity (Path) | WN Purity (WuP) |
|---|---|---|---|---|
| Raw GloVe | 0.4922 | 0.9723 | 0.1158 | 0.3571 |
| VNR | **0.4083** | 0.9677 | **0.1171** | **0.3579** |
| L2-normalized | 7.2403 | 0.9640 | 0.1166 | 0.3563 |
| ZCA-whitened | 0.6714 | **0.9610** | 0.1113 | 0.3412 |

