# OpenReview forum: "Disentangling Superpositions: Interpretable Brain Encoding Model with Sparse Concept Atoms"
_NeurIPS.cc/2025/Conference — NeurIPS 2025 poster_

### Official Review · Reviewer_buH2 · 2025-06-15

**Clarity:** 3
**Significance:** 2
**Originality:** 2
**Rating:** 4
**Confidence:** 2

**Summary:**

The paper proposes Sparse Concept Encoding Model in order to the interpretability of the encoding models for predicting brain activity. While dense feature models are accurate, they lack interpretability because feature representations of different concepts can overlap due to superposition. Their model transforms dense embeddings into a high-dimensional, sparse space composed of learned, interpretable concept atoms. The model matches the performance of conventional dense models on fMRI data, while enabling interpretation of features that dense models don't have access to.

**Questions:**

* In Section 3.2, "we observed a subset that activated broadly across many high-frequency words used in divergent contexts." Can you elaborate more on this, either by visualizations or providing logic on why this could happen?
* line 89, "where" should not be in latex mode.
* line 147, which appendix?
* Section 2.4 has a title but with missing content

**Ethical Concerns:**

["NO or VERY MINOR ethics concerns only"]

**Final Justification:**

I would like to thank the authors for their clarification. I would like to raise my score to 4.

**Limitations:**

yes

**Quality:**

3

**Strengths And Weaknesses:**

Strengths:
* The paper is clearly written, with beautiful visualizations
* The method is elegant and simple and seems to work

Weaknesses:
* The novelty of the model needs more justification, since the model seems like a conventional dictionary learning method. If the novelity is not the main selling point, the implications for neuroscience should be elaborated on more.
* "Limitations of Dense encoding models" is a bit verbose.

---

> ### Author Rebuttal · Authors · 2025-07-31
>
> The main criticism from buH2 is inadequate novelty, a concern also raised by ymzV. We believe these concerns may stem from misunderstandings about the core goals and contributions of the paper. Below, we restate the intended contribution and its relevance to the NeurIPS neuroscience community.
>
> Our central contribution is to identify a previously overlooked limitation of brain encoding models that use artificial neural network (ANN) features, demonstrate its empirical consequences, and propose a practical solution that enables principled neuroscientific interpretation. Specifically, we formalize the impact of superposition in ANN-based feature spaces: when the number of latent semantic features exceeds the embedding dimensionality, those features become entangled in correlated directions. Thus, the regression weight vector learned for a voxel is non-identifiable: multiple distinct combinations of semantic features can yield the same predicted brain response. This ambiguity makes it impossible to reliably determine which specific concepts a voxel is tuned to, and ultimately limits the utility of dense models for answering neuroscientific questions. Our proposed framework preserves the predictive utility of ANN-derived features while restoring the interpretability necessary for scientific discovery. The framework can be generalized to a wide range of representations (e.g., LLM embeddings) and recording modalities (e.g., ECoG, MEG, or intracranial recordings).
>
> This limitation is broadly relevant to a growing body of work, including several recent papers published at NeurIPS [1,2,3,11], that uses ANN-based features in linear encoding models to predict brain activity. These works often rely on the implicit assumption that learned weight vectors in dense embedding spaces are directly interpretable. Our paper demonstrates that this assumption is flawed. By formalizing this limitation and providing a principled alternative, our work directly engages with this line of research and contributes to an ongoing discussion in the NeurIPS neuroscience community [4,5,6] about how ANN-based brain models should be interpreted and evaluated.
>
>
> > The novelty of the model needs more justification, since the model seems like a conventional dictionary learning method. If the novelity is not the main selling point, the implications for neuroscience should be elaborated on more.
>
> We acknowledge that the individual techniques employed in our approach, dictionary learning and voxelwise regression, are not themselves novel. However, this is, to our knowledge, the first study to formally define the semantic non-identifiability problem in ANN-based brain models, show how it undermines standard interpretability practices, and offer a widely-applicable solution.
>
> We recognize this framing was insufficiently emphasized in the original submission. In the revision, we will: (1) move summaries of the non-identifiability argument and Bayesian framing from Appendix C to the main text; (2) revise Section 2.3 (Limitations of Dense Encoding Models) to clarify the motivation (the full revised section is included in our response to Reviewer buH2); (3) restore the omitted content of Section 2.4, which critically introduces the Sparse Concept Encoding Model; and (4) update the introduction to explicitly state that our goal is to correct a conceptual flaw in a widely used modeling framework rather than to propose a novel algorithm or report new neuroscientific findings. Together, these changes should serve to more clearly highlight the conceptual insight and neuroscientific significance of this work.
>
>
> > "Limitations of Dense encoding models" is a bit verbose.
>
> We appreciate this feedback. We have revised the second paragraph of Section 2.3 to clarify our core argument.
>
> Revised Section 2.3: Limitations of Dense Encoding Models (second paragraph)
>
> Encoding models based on dense ANN embeddings face a fundamental limitation: semantic directions are entangled due to superposition, making it difficult to reliably determine which concepts a voxel is truly selective for. To illustrate, consider two contrasting cases of feature spaces used in voxelwise encoding models. In the ideal case, the feature space is constructed with an interpretable basis, where each axis corresponds to a distinct semantic concept (Figure 1, right). A voxel’s regression weight vector can then be interpreted directly: each coefficient reflects sensitivity to a specific concept. Even when predictors are correlated, standard techniques such as variance partitioning can disentangle their contributions. This is the interpretability goal enabled by the Sparse Concept Encoding Model.
>
> In contrast, dense embeddings such as GloVe (Figure 1, left) do not provide an interpretable basis. Semantic concepts are encoded by correlated linear combinations of basis directions rather than coordinate axes. When a voxel is jointly tuned to multiple concepts (e.g., “time” and “color”), its weight vector lies between their directions. Cosine-based interpretability methods may misidentify this vector as responding to an unrelated concept (e.g., “space”), leading to incorrect conclusions. These methods implicitly assume that each voxel is tuned to a single semantic direction.However, this assumption is inconsistent with decades of neuroscience findings showing that voxels often respond to multiple distinct features. Dense models obscure this multi-concept tuning, limiting their interpretability.
>
>
> > Section 2.4 has a title but with missing content
>
> Thank you for catching this blunder. To our deepest regret, we mistakenly submitted a non-final version of the manuscript, which included several severe organizational blunders and formatting oversights, including the omission of Section 2.4, inconsistent figure references, and missing citations (ymzV, buH2). We sincerely apologize for any confusion this may have caused and appreciate the reviewers’ patience and attention to detail. We are committed to carefully addressing all these issues, and if the paper is accepted, we will make every effort to ensure that the camera-ready version is error-free. We feel certain these revisions will significantly improve the clarity and impact of the paper. The corrected version of Section 2.4 is included below.
>
> Section 2.4: Sparse Concept Encoding Model
>
> To resolve the identifiability limitations of dense embeddings, we construct a sparse, overcomplete semantic space. Specifically, we expand the original d-dimensional embedding space into an m-dimensional space ( m>d) using sparse dictionary learning. Each axis in this expanded space corresponds to a distinct semantic direction, which we term a concept atom. These atoms are learned from a large text corpus using non-negative sparse coding (Fig. 2b). After dictionary learning, each stimulus word is re-encoded as a sparse, non-negative linear combination of concept atoms. This sparse feature matrix replaces the dense feature matrix in the voxelwise encoding model pipeline (Fig. 2c). Because the dictionary is overcomplete and approximately orthogonal, the resulting Gram matrix G=I. This structure ensures each voxel’s weight vector provides a direct ,interpretable readout of its tuning to individual concept atoms. The non-negativity constraint in dictionary learning ensures interpretability: positive weights reflect tuning to specific concepts, while negative weights imply suppression. Crucially, since each concept atom occupies its own axis in the new space, voxels jointly selective for multiple semantic features can be cleanly disentangled. This model thus enables unambiguous, voxel-level interpretation of conceptual tuning for each voxel.
>
> > In Section 3.2, "we observed a subset that activated broadly across many high-frequency words used in divergent contexts." Can you elaborate more on this…
>
> In dense embeddings like GloVe or FastText, high-frequency words tend to lie in similar directions—not due to shared semantics, but because of frequency-related statistical patterns [12]. High-frequency words also tend to lie closer to the origin, likely because their embedding vectors represent the average over more diverse contexts [13]. Sparse coding picks up on these shared directions, yielding atoms that broadly activate across semantically diverse but frequent words.
>
> References
>
> [1] Huth, A. G., De Heer, W. A., Griffiths, T. L., Theunissen, F. E., & Gallant, J. L. (2016). Natural speech reveals the semantic maps that tile human cerebral cortex. Nature, 532(7600), 453-458.
>
> [2] Tang, J., Du, M., Vo, V., Lal, V., & Huth, A. (2023). Brain encoding models based on multimodal transformers can transfer across language and vision. Advances in Neural Information Processing Systems, 36, 29654-29666.
>
> [3] Wang, A., Tarr, M., & Wehbe, L. (2019). Neural taskonomy: Inferring the similarity of task-derived representations from brain activity. Advances in neural information processing systems, 32.
>
> [4] Schaeffer, R., Khona, M., Chandra, S., Ostrow, M., Miranda, B., & Koyejo, S. (2024, October). Position: maximizing neural regression scores may not identify good models of the brain. In UniReps: 2nd Edition of the Workshop on Unifying Representations in Neural Models.
>
> [5] Canatar, A., Feather, J., Wakhloo, A., & Chung, S. (2023). A spectral theory of neural prediction and alignment. Advances in Neural Information Processing Systems, 36, 47052-47080.
>
> [6] Ostrow, M., Eisen, A., Kozachkov, L., & Fiete, I. (2023). Beyond geometry: Comparing the temporal structure of computation in neural circuits with dynamical similarity analysis. Advances in Neural Information Processing Systems, 36, 33824-33837.
>
> [11] Wehbe, L., Huth, A. G., Deniz, F., Gao, J., Kieseler, M. L., & Gallant, J. L. (2016). BOLD predictions: Automated simulation of fMRI experiments. NeurIPS Demonstr. Track.

---

> > ### Comment · Reviewer_buH2 · 2025-08-08
> >
> > I would like to thank the authors for their clarification. I would like to raise my score to 4.

---

> > > ### Author Response · Authors · 2025-08-09
> > >
> > > Thank you again for the insightful review! We particularly appreciate the reviewer pointing out section 2.3 was verbose. The revisions we made based on the reviewer's feedback surely serve to strength the paper.

---

### Official Review · Reviewer_iQoW · 2025-06-27

**Clarity:** 3
**Significance:** 2
**Originality:** 3
**Rating:** 5
**Confidence:** 4

**Summary:**

The goal of the study is to develop more interpretable fMRI encoding models. The authors provided mathematical and empirical demonstrations that dense embeddings lead to ambiguities in model weight interpretation. The authors then developed a theoretically motivated approach that decomposes the dense embeddings into sparse combinations of concept atoms. The authors estimated encoding models using the sparse coefficient vectors as stimulus features and demonstrated that sparse models achieve similar predictive performance to dense models. Finally the authors analyzed the sparse model weights to visualize how concepts are encoded across the cortex.

**Questions:**

In 3.5 could it be the case that the sparse model predicts more overlap for all concepts (rather than just time, space, and number concepts)? As a control could you compare the sparse and dense models on a different set of concepts that are actually hypothesized to be separate in the brain?

The interpretations will depend on the actual atoms that are chosen for each concept. In 3.5 the authors chose one representative atom for each concept (time, space, and number) and said that the others atoms provided similar results (line 534). Is there a way to quantify this similarity?

The overall prediction performance is similar but are there any brain areas where prediction performance differs between the dense and sparse models?

**Ethical Concerns:**

["NO or VERY MINOR ethics concerns only"]

**Final Justification:**

The authors have adequately answered my questions about the study. They have also provided thoughtful responses to the other reviewers that clarify the motivation of the study. I think that this is an interesting study and will keep my score at 5.

**Limitations:**

Yes

**Quality:**

3

**Strengths And Weaknesses:**

Strengths

The sparse concept encoding model is clearly motivated and the authors provided a convincing argument that sparse models are more interpretable than dense models. The sparse models were able to attain comparable performance to dense models. The vector norm reparameterization technique for differentiating between high frequency words could be useful for future studies. The case studies demonstrate how sparse models can be used to address long standing questions in cognitive neuroscience.

Weaknesses

The main benefit of the sparse model is interpretability but this can be difficult to evaluate. For instance the authors show that the dense model predicts more overlap across time, space, and number concepts than the sparse model, and they claim that this finding is more consistent with known behavioral and psychophysical results. However it would also be helpful to evaluate which model is more consistent with known neuroimaging results. It appears that the Huth et al. data contains separate ROIs, is there a way to compare the dense and sparse models to the ROIs?

---

> ### Author Rebuttal · Authors · 2025-07-31
>
> We sincerely thank Reviewer iQoW for their thoughtful and constructive review. We especially appreciate their insightful questions based on a clear theoretical understanding of the work and methods therein. We hope that our responses to these questions provide additional helpful context for the panel’s overall evaluation.
>
> > The main benefit of the sparse model is interpretability but this can be difficult to evaluate. For instance the authors show that the dense model predicts more overlap across time, space, and number concepts than the sparse model, and they claim that this finding is more consistent with known behavioral and psychophysical results. However it would also be helpful to evaluate which model is more consistent with known neuroimaging results. It appears that the Huth et al. data contains separate ROIs, is there a way to compare the dense and sparse models to the ROIs?
>
> We thank the reviewer for this thoughtful suggestion. The Huth et al. dataset includes standard regions of interest (ROIs) localized using widely adopted functional localizers. These include visual category ROIs, auditory cortex ROIs and motor ROIs. These ROIs are well-established for sensory and motor functions. However, there are no standardized ROIs for abstract concepts such as time, space, or family. While some prior studies have implicated the intraparietal sulcus (IPS) in number and spatial processing, and medial prefrontal and parietal cortices in social or autobiographical concepts, these findings are not consistently replicable. This reflects a broader challenge in cognitive neuroscience. Traditional task-based fMRI studies often collect ~10 minutes of data per subject using isolated stimuli and analyze group-level averages. This approach has faced reproducibility issues, as results often fail to generalize across datasets or paradigms. By contrast, the Huth et al. dataset used in our study adopts a naturalistic paradigm, collecting multiple hours of rich, continuous story-listening data per participant. Their data reveal more distributed and overlapping cortical representations of conceptual content, but also makes ROI-based validation more difficult. Our work is designed to address this challenge. Our proposed Sparse Concept Encoding Model enables interpretable voxel-level tuning even for abstract concepts that lack well-defined ROIs.
>
> In the revision, we will clarify that most abstract concepts do not have standardized ROIs. We will also reference prior neuroimaging findings on the cortical representation of number, time, and space, and include qualitative comparisons with the corresponding maps produced by our models.
>
> > In 3.5 could it be the case that the sparse model predicts more overlap for all concepts (rather than just time, space, and number concepts)? As a control could you compare the sparse and dense models on a different set of concepts that are actually hypothesized to be separate in the brain?
>
> This is an excellent question! As we explain in the paper, dense embedding models lack a principled method for generating concept-level cortical maps. To enable rough comparison, Appendix F reports heuristic maps for the dense encoding model. These are not ground-truth concept maps, but approximations constructed using heuristic methods. Prior work has used several such heuristics, including:
>
> * Cosine similarity: assigning each voxel the concept whose direction is nearest to its weight vector [2,9,10]. This often underestimates overlap, as mixed selectivity is ignored.
>
> * Projection: projecting voxel weights onto semantic directions of interest [1,11]. This often overestimates overlap due to spurious correlations caused by superposition.
>
> In our paper, we adopt a middle-ground heuristic, labeling a voxel as selective for a concept if that concept appears in its top-10 cosine similarity matches. If we use top-1, overlap becomes minimal; with top-20, it increases. This sensitivity illustrates why concept overlap under dense models is heuristic-dependent and not robust. In revision, we will make this clearer in the main text and can include additional comparisons using different thresholds in the appendix if reviewers would find this helpful.
>
> > The overall prediction performance is similar but are there any brain areas where prediction performance differs between the dense and sparse models?
>
> Thank you for this helpful question! It can best be answered in the form of a flatmap showing the difference in prediction performance between the two models across the cortical surface, similar to Figure 9 in Deniz et al [9]. While NeurIPS rebuttal guidelines do not permit inclusion of figures, we have conducted this analysis and found no consistent spatial pattern indicating one model outperforms the other across specific cortical regions. In the revision, we will include this differential performance map in the appendix to support this observation.
>
> > The interpretations will depend on the actual atoms that are chosen for each concept. In 3.5 the authors chose one representative atom for each concept (time, space, and number) and said that the others atoms provided similar results (line 534). Is there a way to quantify this similarity?
>
> We can quantify the similarities by calculating the cosine similarities of cortical maps for atoms within each concept group (e.g. “time - duration” and “time - moments”), and compare these values with the distribution of cosine similarities for all possible pairs of atoms. In revision we will report the statistics.
>
> References
>
> [1] Huth, A. G., De Heer, W. A., Griffiths, T. L., Theunissen, F. E., & Gallant, J. L. (2016). Natural speech reveals the semantic maps that tile human cerebral cortex. Nature, 532(7600), 453-458.
>
> [2] Tang, J., Du, M., Vo, V., Lal, V., & Huth, A. (2023). Brain encoding models based on multimodal transformers can transfer across language and vision. Advances in Neural Information Processing Systems, 36, 29654-29666.
>
> [9] Deniz, F., Nunez-Elizalde, A. O., Huth, A. G., & Gallant, J. L. (2019). The representation of semantic information across human cerebral cortex during listening versus reading is invariant to stimulus modality. Journal of Neuroscience, 39(39), 7722-7736.
>
> [10] Meschke, E. X., Castello, M. V. D. O., Tour, T. D. L., & Gallant, J. L. (2023). Model connectivity: leveraging the power of encoding models to overcome the limitations of functional connectivity. BioRxiv, 2023-07.
>
> [11] Wehbe, L., Huth, A. G., Deniz, F., Gao, J., Kieseler, M. L., & Gallant, J. L. (2016). BOLD predictions: Automated simulation of fMRI experiments. NeurIPS Demonstr. Track.

---

> > ### Comment · Reviewer_iQoW · 2025-08-04
> >
> > Thank you for your response! I think that including more dense encoding model comparisons with different heuristics in the appendix will help clarify the strengths of the sparse encoding model. I think that this is an interesting study and will still recommend "Accept".

---

> > > ### Author Response · Authors · 2025-08-09
> > >
> > > Thank you again for your thoughtful suggestions! We will include the analyses you recommended and are confident it will help improve the paper.

---

### Official Review · Reviewer_ZoiN · 2025-06-30

**Clarity:** 3
**Significance:** 4
**Originality:** 3
**Rating:** 5
**Confidence:** 4

**Summary:**

The paper presents e Sparse Concept Encoding Model, a novel method to perform encoding analysis in fMRI data. The contribution lies in using dictionary learning to decompose dense word embeddings (GloVE) into a sparse set of quasi-orthoghonal concept atoms.

Thanks to such decomposition, the weight vectors resulting of the encoding linear model become directly interpretable. In such way, SCEM provides with key insights about where in the brain different concepts are encoded across patients.

The results show that space, time and number representations are encoded in the same brain regions,hinting towards a unified magnitude system. The paper also shows a consistent anatomical mapping of different concepts (other than space, time and number) across patients.

**Questions:**

What is the interpretation of blue vs red in Figure 5? It seems like the brain regions are consistent across patients  but not the colors.

**Ethical Concerns:**

["NO or VERY MINOR ethics concerns only"]

**Limitations:**

yes

**Quality:**

4

**Strengths And Weaknesses:**

## Strengths

- The paper is well written, the motivation, aim, methods and results are well conveyed.
- The method provides with a principled way to better interpret encoding models
- The results look consistent and coherent with previous work

## Weaknesses

- It is not very clear to me if such "unified magnitude system" can be concluded from figure 6. While there is quite a bit of overlap (white), the maps look a bit disordered

---

> ### Author Rebuttal · Authors · 2025-07-31
>
> We sincerely thank Reviewer ZoiN for their thoughtful review and for highlighting several key strengths of our work. We are especially grateful for their recognition of the principled nature of our method and the clarity of our presentation.
>
> > It is not very clear to me if such "unified magnitude system" can be concluded from figure 6. While there is quite a bit of overlap (white), the maps look a bit disordered
>
> We appreciate this observation. We agree that the maps in Figure 6 appear noisy. However, this level of variability is expected in within-subject functional maps generated from VEM studies using naturalistic stimuli [1,7, 9,10]. As a non-invasive imaging technique, fMRI inherently suffers from low signal-to-noise ratios. Group-level analyses, which project participant-specific weights to a common template and averaging across individuals, typically yield smoother and more structured maps. However, this comes at the cost of reduced sensitivity to individual variability and can obscure meaningful discoveries [7]. As noted in the rebuttal to ymzV, we are happy to include group-level maps in the revision if the reviewers believe this would strengthen the evaluation of the results.
>
> > What is the interpretation of blue vs red in Figure 5? It seems like the brain regions are consistent across patients but not the colors.
>
> We thank the reviewer for this question. In Figure 5, red indicates positive tuning and blue indicates negative tuning to a given concept atom. That is, red voxels show increased predicted responses when the concept is present in the stimulus, while blue voxels show decreased predicted responses (relative to the mean, since responses have been mean-centered for each voxel). In our revision, we will revised Sections 2.3 and 2.4 to make this interpretation more clear. Please see the revisions in our rebuttal to Reviewer buH2.
>
> Regarding the apparent color inconsistency across participants, much of this inconsistency is expected from individual differences in cortical anatomy and functional organization. The 2D flatmaps are created by flatting each subject’s cortical surface through a manual cutting procedure [1,7], which introduces further variations in spatial alignment. If one examines the inflated cortical surface renderings below the flatmaps in Figure 5, the spatial correspondence of activated regions across participants might be more evident.
>
> References
>
> [1] Huth, A. G., De Heer, W. A., Griffiths, T. L., Theunissen, F. E., & Gallant, J. L. (2016). Natural speech reveals the semantic maps that tile human cerebral cortex. Nature, 532(7600), 453-458.
>
> [7] Dupré la Tour, T., Visconti di Oleggio Castello, M., & Gallant, J. L. (2025). The Voxelwise Encoding Model framework: a tutorial introduction to fitting encoding models to fMRI data. Imaging Neuroscience, 3, imag_a_00575.
>
> [9] Deniz, F., Nunez-Elizalde, A. O., Huth, A. G., & Gallant, J. L. (2019). The representation of semantic information across human cerebral cortex during listening versus reading is invariant to stimulus modality. Journal of Neuroscience, 39(39), 7722-7736.
>
> [10] Meschke, E. X., Castello, M. V. D. O., Tour, T. D. L., & Gallant, J. L. (2023). Model connectivity: leveraging the power of encoding models to overcome the limitations of functional connectivity. BioRxiv, 2023-07.

---

> > ### Comment · Reviewer_ZoiN · 2025-08-08
> >
> > The authors have addressed most of my concerns, I will keep my score and recommend acceptance

---

> > > ### Author Response · Authors · 2025-08-09
> > >
> > > We thank Reviewer ZoiN again for their incisive review! We are greatly encouraged by their enthusiasm towards this work.

---

### Official Review · Reviewer_ymzV · 2025-07-03

**Clarity:** 2
**Significance:** 2
**Originality:** 2
**Rating:** 3
**Confidence:** 4

**Summary:**

1. The authors learn a mapping from 300-D GloVe word embeddings to 1000-D sparse vectors, each representing coefficients of an overcomplete basis for the embedding space, via a dictionary learning framework. The basis vectors can be intepreted as semantic concepts by retrieving words assigned a large coefficient for that vector.

2. They then regress fMRI recordings from a listening task using the sparse vectors as input and test on held-out stories.

3. They find that (i) sparse embeddings show similar brain encoding performance to dense embeddings while offering more intepretability, (ii) the regression weight vectors corresponding to select concepts, when projected on to the brain, are visually similar across two subjects, and (iii) there is visual overlap between the projections corresponding to "time", "space", and "number" concepts.

**Questions:**

None.

**Ethical Concerns:**

["NO or VERY MINOR ethics concerns only"]

**Final Justification:**

The authors have improved their empirical evaluation as suggested, so I will raise my score to 3 (Borderline reject).

I am not willing to go higher after the rebuttal since they:
1. Overstate the contribution of a ''first mathematical framework'' while the paper contains no mathematical or theoretical novelty.
2. Claim that dense embeddings *only enable* heuristic analysis while not substantiating this adequately (e.g., by trying a bunch of dense-embedding based methods and showing that they all fall short).
3. Only report experiments with GloVe even when highlighted as a major limitation for interpretability. The authors point out that sparsifying an LLM is non-trivial, but I never suggested that the LLM itself be sparsified, just that the chosen dictionary learning approach be applied on top of LLM-based embeddings instead of non-contextual embeddings. I believe this is simple to implement by drawing on prior work and it seems the authors have severely misrepresented my concern.

**Limitations:**

Yes.

**Paper Formatting Concerns:**

None.

**Quality:**

2

**Strengths And Weaknesses:**

Strengths:
1. The approach can inspire future work on gaining a deeper understanding of brain activity based on known semantic concepts in the stimulus.

Weaknesses:
1. Lack of novelty: None of the methods in the paper are novel. Sparse coding is a well-known approach to extract interpretable features from neural network layers, including LLMs (as described in Section 2.2). The specific dictionary learning framework used was introduced in previous work (as mentioned in line 163). The resulting regression analysis of fMRI recordings is also standard.

2. Lack of rigor:\
**a)** In lines 159-160, the authors say they leave validation on more participants to future work and present only "exploratory findings", which falls short of typical NeurIPS empirical standards. While the exploratory results may serve as qualitative examples, it is crucial to present a more robust quantitative analysis, which includes (i) clearly defining evaluation metrics, (ii) utilizing data from all (seven) participants, and (iii) reporting the variance in the results across participants whenever possible. Moreover, this all needs to be included in the main paper, not pushed to the appendix.\
**b)** Section 3.4 claims inter-subject consistency of per-concept cortical maps obtained from the learned weight matrix when using sparse embeddings. However, the only evidence for this is a visualization (Figure 5) for two selected concepts and two participants. The same applies to the claim of increased inter-subject variability (line 222) when using dense embeddings. The authors should instead quantify the variability across subjects (aggregated over concepts) when using sparse vs. dense embeddings to provide more tangible and reliable evidence for their claims. They could possibly also examine brain regions where the variability is more pronounced using either method.\
**c)** Section 3.5 attempts to show overlap in the neurons responding to "time", "space", and "number" concepts. Again, the authors have not quantified the apparent overlap, instead resorting to a visualization (Figure 6). A straightforward solution is to report some kind of intersection over union (IoU) of the neurons responding to the three concepts.\
**d)** Section 3.6 mentions "similar clustering structure" (line 253), "correlations were uniformly low" (line 257), and "moderate positive correlations" (line 260). None of these are quantified in the paper.\
**e)** Lines 186-187 state that Vector Norm Reparameterization qualitatively leads to "sharper" concepts. However, no evidence is provided for this claim in the main paper. There is also no derivation for the form of the rescaling in the appendix.

3. Section 2.4, presumably meant to describe the method, has no content.

4. There are no references to the literature in (i) lines 47-48 for the claim that each neuron in the brain likely responds to multiple semantic concepts and (ii) lines 182-183 for the claim that the embeddings for high-frequency words tend to cluster near the origin.

5. In lines 61-63, the authors claim to present the first mathematical framework showing that several embedding inputs may be mapped to the same predicted brain response. However, this is an expected result in multi-output linear regression since an exact one-to-one map would require the rank of the learned regression weight matrix to match the number of columns, which is not guaranteed in practice.

6. The paper uses GloVe vectors, which provide non-contextual representations of words, while participants hear words in context. This represents a lack of consistency between the inputs and outputs of the regression. While acknowledged as a limitation in lines 272-276, I do not see a reason for not using LLM-based embeddings in the analysis given their ease of access (e.g., through Hugging Face) and the sheer number of open-source LLMs available in 2025.

7. Section 3.1 does not seem to add much value to the paper as it is already known that dictionary learning can produce basis vectors capturing interpretable concepts, as explained in Section 2.2. There is no need to show "consistency with prior findings" (line 168) since this section is simply replicating prior findings. A simple visualization would suffice to emphasize this point.

8. The intention of Section 3.6 is not clear. It starts (lines 241-244) by wanting to investigate the similarity between brain representations of different abstract concepts. However, there is no reference to these "abstract concepts" anywhere else in the section. It is not clear how "the top 20 concept atoms" (line 247) analyzed are linked with the abstract concepts.

9. The authors state that a limitation of dense embedding-based brain encoding is that each component contains a combination of multiple semantic features, rendering the results less interpretable. However, lines 47-48 and 142-143 imply that voxels behave in a similar manner, i.e., a single voxel may respond to multiple concepts. The paper lacks any discussion of this – and its effect on interpretable brain encoding – despite the emphasis on interpretability throughout the paper.

10. Minor:\
**a)** "where" in line 89 is in math mode.\
**b)** "Fig 2a" => "Fig 3a" (line 131).\
**c)** "the the" => "the" (line 152).\
**d)** "Figure 3" => "Figure 2" (line 169).\
**e)** "Appendix D" => "Appendix F" (line 223).\
**f)** "Section 3.5" => "Section 3.4" (line 248).\
**g)** There is no reference to Figure 5 in Section 3.4.\
**h)** There is no reference to Figure 6 in Section 3.5.\
**i)** "Subiect" => "Subject" (Figure 5b).\
**j)** "ATOM 81" => "ATOM 997" (Figure 6c).\
**k)** "Space" and "Time" colors are flipped in the Venn diagram in Figure 6.

---

> ### Author Rebuttal · Authors · 2025-07-31
>
> We thank reviewer ymzV for the detailed evaluation, thoughtful comments, and meticulous feedback on technical and editorial issues. We have carefully considered all critiques and are confident we can address them fully and that revision will substantially strengthen the work.
>
> While we agree with most of the reviewer’s concerns, we respectfully disagree with two key criticisms: (1) inadequate novelty, and (2) insufficient number of subjects and cross-participant analysis. For brevity, we will respond in detail to (1) in our rebuttal to buH2 and (2) here:
>
> > …the authors say they leave validation on more participants to future work and present only "exploratory findings", which falls short of typical NeurIPS empirical standards...
>
> We appreciate the reviewer’s emphasis on robust quantitative analysis. However, this concern may partially reflect a misunderstanding of the modeling framework adopted.
>
> Our analysis adopts the Voxelwise Encoding Model (VEM) framework [7], which emphasizes high-resolution, within-subject modeling using large volumes of data per participant. VEM differs from traditional group-level fMRI designs, which typically entail collecting short recordings (around 20 minutes) from many participants (≥ 20) and evaluating hypotheses by averaging voxel responses across individuals. These traditional approaches lack held-out test evaluation and rely heavily on point-null hypothesis testing, which has been shown to suffer from reproducibility issues. The VEM framework by contrast, entails extensive data collection (around 3 hours) from each participant to enable voxelwise regression models to be fit independently in each subject’s native cortical space. Model performance is evaluated on held-out test data. Thus, each participant serves as a full replication of the hypothesis test. Group-level prevalence can thus be assessed by projecting subject-specific model parameters to a common template surface.
>
> Given the limited conference format, we prioritized within-subject analyses. These analyses adhere to best practices in data science: model weights are estimated using cross-validation on a training dataset; predictive performance is evaluated on held-out test data; confidence intervals of model weights are computed via bootstrapping; and statistical significance of prediction is assessed through nonparametric permutation testing. Detailed methods are in Appendix A. rather than the main text because the goal of this paper is to introduce a broadly applicable modeling framework rather than to present new neuroscientific discoveries. That said, we agree group-level results can provide valuable complementary evidence. In the revision, we are happy to include analyses from all seven participants and report cross-subject consistency and variability in model performance and cortical maps if the reviewers feel this is essential.
>
> > … authors claim to present the first mathematical framework showing several embedding inputs may be mapped to the same predicted brain response... this is an expected result in multi-output linear regression since exact one-to-one map would require the rank of the learned regression weight matrix to match the number of columns, …
>
> We believe this comment may reflect a misunderstanding of our central claim. The reviewer references “multi-output linear regression” and “rank of the regression weight matrix”. However, our argument pertains to single-output linear regression, which is standard in voxelwise encoding models. We also do not argue that underdetermined regression problems have multiple solutions. This is a well-known fact often addressed by regularization techniques such as ridge regression. Rather, we highlight a more subtle issue: even when the regression problem is well-posed and the solution is unique, the interpretation of that solution becomes ambiguous when dense embeddings are used. Because semantic directions in dense spaces are often correlated due to superposition, a single weight vector can correspond to multiple plausible combinations of underlying semantic features. In other words, the solution is mathematically unique but semantically non-identifiable.
>
> Although the mathematical proof is straightforward, this interpretability problem has been largely overlooked. In prior studies, weight vectors from dense encoding models were assumed to reflect meaningful voxel tuning, and standard practices such as nearest-neighbor word retrieval were used to interpret them. To our knowledge, we are the first to identify and formalize this problem. In the revision, we will clarify the scope of our argument and emphasize that our contribution is not to present a novel theoretical result, but to identify and formalize a conceptual flaw in a widely used modeling practice.
>
> > The authors state a limitation of dense embedding-based brain encoding is that each component contains a combination of multiple semantic features…
>
> We appreciate this comment and agree that clarifying how voxel multi-selectivity affects interpretability is central to our work. In fact, our proposed sparse framework is specifically designed to address this issue. The limitation of dense embeddings is not that voxels respond to multiple concepts but that dense models fail to disentangle the contributions of each concept due to the superposition of semantic directions. To make this connection clearer, we have revised Section 2.3 (Limitations of Dense Encoding Models) and restored the missing content of Section 2.4, which introduces the Sparse Concept Encoding Model. Both revised sections are included in our response to Reviewer buH2. We deeply apologize for the confusion caused by the omission.
>
> > The paper uses GloVe vectors, which provide non-contextual representations of words…
>
> We chose GloVe for two main reasons. First, our goal was to isolate and examine the interpretability gap between dense and sparse encoding models in direct comparison with prior work. Word embeddings such as GloVe have been widely used in previous voxelwise encoding model studies, making them a natural baseline. Second, contextual embeddings introduce additional modeling decisions (e.g., contextual window length) that would distract from the core contribution of our paper: identifying and resolving the interpretability problem in dense-to-sparse encoding. That said, we fully agree that applying our method to LLM-based embeddings is a natural next step. In fact, we are currently working on extending this work using gemma-2. If the reviewers believe including discussions of contextual embeddings is crucial to the acceptance of this work, we are happy to include these in the revision.
>
> > …authors have not quantified the apparent overlap…solution is to report some kind of intersection over union (IoU) of neurons…
>
> > Section 3.6 mentions "similar clustering structure"...None of these are quantified...
>
> We truly appreciate these suggestions and agree these sections should include quantitative descriptions in addition to visualization. In revision we will include textual descriptions of these statistics.
>
> > Section 3.4 claims inter-subject consistency of per-concept cortical maps obtained from the learned weight matrix when using sparse embeddings…
>
> We appreciate the reviewer’s suggestion. As our study adopts the VEM framework, our main focus has been on within-subject modeling. While we do present qualitative evidence of cross-subject consistency in Figure 5, we acknowledge that the current manuscript lacks a quantitative group-level analysis. That said, inter-subject consistency is not a central claim of our work. Moreover, for the dense encoding model, the comparison maps are derived from heuristic procedures and vary substantially depending on the method used, as is discussed in rebuttal to buH2. Nonetheless, we agree that adding group-level metrics could strengthen the manuscript. If the reviewers believe this is necessary for acceptance, we are happy to fit models on all seven participants and report inter-subject similarity metrics in the revision.
>
> > Section 3.1 does not seem to add much value to the paper as it is already known that dictionary learning can produce basis vectors capturing interpretable concepts…
>
> We appreciate this suggestion and agree Section 3.1 should be condensed. One of the original goals of the section was to show that many semantic directions in dense embeddings are moderately correlated, which motivates the need for a disentangled feature space. In revision we will streamline this section.
>
> > The intention of Section 3.6 is not clear…
>
> We thank the reviewer for highlighting this lack of clarity. We agree the document did not sufficiently explain the link between these categories and the selected concept atoms. To clarify: each learned concept atom corresponds to a distinct semantic dimension, identified by inspecting the top stimulus words associated with that atom (see Appendix Table 1). These atoms can be interpreted as representing abstract concepts, such as “oppositional conjunctions” or “body positions.” In Section 3.6, we focused on the 20 atoms with the highest average cortical activation, as they provide the most robust signal for cross-atom comparisons. In revision we will clarify the connection between concept atoms and abstract categories. Specifically, we will clearly state that the selected atoms correspond to interpretable abstract concepts; reference Appendix Table 1, which lists the top 50 atoms; and move Figure 7 (the labeled dendrogram and clustered similarity matrix) into the main text.
> > Lines…state Vector Norm Reparameterization qualitatively leads to "sharper" concepts. However, no evidence is provided…
>
> In the revision, we can include a table of subjective assessments demonstrating that Vector Norm Reparameterization indeed yields more semantically distinct concepts, as well as a figure which intuitively demonstrates the way this rescaling works.

---

> ### Comment · Reviewer_ymzV · 2025-08-06
>
> The authors agree with most of my concerns and have partially responded to others. My main concern on lack of rigor has not been addressed in the rebuttal, which provides no quantitative metrics in response to any of the points highlighted in my initial review. **I will therefore keep my rating of 2 (Reject).**
>
> See below for specifics:
> 1. The authors state that demonstrating the ''empirical consequences'' of existing neural network-based encoding models adds novelty to the work. However, neither the paper nor the rebuttal presents any reliable empirical evaluation. All the evidence presented is in the form of visualizations.
>
> 2. Lack of rigor:\
> **a)** Even if within-subject analysis is standard practice for the VEM framework, this analysis should be performed for all seven subjects. Otherwise, it is not clear how the authors selected the two participants, which may raise suspicion that they only presented the most favorable results. It is thus crucial to also report the variance across participants.\
> **b)** In the rebuttal, the authors state inter-subject consistency is not a central claim. However, they still make this claim in line 218 with no quantitative evidence to support it. This and the heuristic analysis with dense embeddings further highlight the lack of strong evaluation in the paper.\
> **c)** The authors agree neuron overlap should be quantified but have not done so in the rebuttal. This would have involved minimal extra work since it requires no new experiments, just reporting standard metrics.\
> **d)** The authors agree ''clustering structure'' and correlations should be quantified but have not done so in the rebuttal. In particular, describing correlations as ''uniformly low'' or ''moderately positive'' without reporting the actual numbers is not acceptable in my view.
>
> 3. The authors promise to restore the omitted content in Section 2.4, which they say ''critically introduces'' their model. The fact that this section is important to the story (and in supposedly addressing my interpretability concern in point 9) but is completely missing from the submission is a clear sign that the paper requires major revision and is not ready for publication as things stand.
>
> 4. Comment on references to literature not addressed.
>
> 5. If the intention is to simply highlight an interpretability issue with dense embeddings, the authors' claim of presenting a ''first mathematical framework'' is an overstatement, not a misinterpretation on my part.
>
> 6. I am not satisfied with the reasons for using GloVe embeddings. The authors emphasize interpretability in their rebuttal, but non-contextual embeddings clearly detract from this goal by entangling all possible meanings of a word into the same vector (while participants interpret with a specific meaning). Regarding additional modeling decisions, the authors could take inspiration from the abundance of existing work using contextual embeddings for brain encoding. In my view, using LLM-based embeddings is not merely a ''natural next step'', but a crucial missing piece with respect to the intent of this paper.

---

> ### Author Response · Authors · 2025-08-08
>
> Due to length limits, we had prioritized addressing core questions over listing extensive metrics. We now resolve the main concern on “lack of rigor” with explicit numerical data & address remaining critiques.
>
> > a) … all 7 subjects… b) … authors should instead quantify the variability across subjects...
>
> We ran all analyses across 7 subjects and now report inter-subject consistency of per-concept cortical maps using mean pairwise cosine similarity after projecting subject weight maps to fsaverage space. Across the 20 concept atoms, sparse embeddings yielded higher inter-subject consistency (mean ± SD: 0.26 ± 0.04) than dense embeddings (0.09 ± 0.04), both overall & per factor. We will include a line plot of per-factor consistency.
>
> > c) ...neuron overlap should be quantified but have not done so…
>
> We report IoU values for voxels selective to time, space, & number across subjects. Averaged over 7 subjects, IoU was 0.34 ± 0.05 for time–space, 0.32 ± 0.07 for time–number, 0.32 ± 0.07 for space–number, & 0.18 ± 0.05 for voxels selective to all three.
>
> > d) Section 3.6 mentions "similar clustering structure"...None of these are quantified...
>
> Revised:
>
> The 20×20 concept-level similarity matrices showed consistent clustering across all 7 subjects (mean off-diagonal Pearson r = 0.70 ± 0.08). To confirm this structure was not inherited from the input features, we computed pairwise correlations between sparse feature vectors. Across 190 unique factor pairs, absolute Pearson correlations were uniformly low (|r| = 0.00 ± 0.10; 85% < 0.10), confirming SCEM effectively disentangles semantic directions. In contrast, cosine similarities between the same factors in dense GloVe space showed moderate positive overlap (0.16 ± 0.14; 87% > 0), suggesting dense embeddings obscure fine distinctions between semantic directions.
>
> Cross-Subject Similarity Matrix
>
> ||S01|S02|S03|S04|S05|S06|
> |-|-|-|-|-|-|-|
> |S02|.78||||||
> |S03|.79|.79|||||
> |S04|.75|.77|.71||||
> |S05|.58|.61|.59|.65|||
> |S06|.73|.72|.76|.72|.50||
> |S07|.71|.77|.72|.71|.61|.63|
>
> > …the authors' claim of presenting a ''first mathematical framework'' is an overstatement, not a misinterpretation on my part
>
> This is indeed the first mathematical framework to formalize the interpretability flaw. Since the formal proof is simple, we stated “In the revision, we will clarify the scope of our argument & emphasize our contribution is not to present a novel theoretical result, but to identify & formalize a conceptual flaw in a widely used modeling practice.”
>
> ymzV states “...not a misinterpretation on my part”, but this is not the misinterpretation we highlighted in rebuttal. We suggested ymzV misunderstood our central argument due to discrepancies in their review: they referenced “multi-output linear regression” & “rank of the regression weight matrix”, but our argument is about neither.
>
> > ...heuristic analysis with dense embeddings further highlight the lack of strong evaluation in the paper.
>
> This comment also suggests a misreading. That dense embeddings only enabling flawed heuristic analysis is the exact problem we highlight and rectify. Showing this limitation for comparison does not “highlight the lack of strong evaluation in the paper”.
>
> > ..references to literature not addressed…
>
> (i) Olman (2023); Rigotti et al. (2013); Huth et al. (2016)
> (ii) Arora et al. (2016); Schakel & Wilson (2015)
>
> > The authors promise to restore the omitted content in Section 2.4…is a clear sign the paper … is not ready for publication
>
> We respectfully disagree the paper is not publication-ready on this basis. As stated in rebuttal: “we have restored the missing content of Section 2.4...Both revised sections are included in our response to buH2.”  We expressed deep regret for mistakenly submitting a non-final manuscript, but we believe the revised content provided is publication-ready.
>
> > I am not satisfied with the reasons for using GloVe embeddings…using LLM-based embeddings is not merely a ''natural next step'', but a crucial missing piece with respect to the intent of this paper.
>
> While LLM can improve models by incorporating context, this improvement is largely orthogonal to our paper’s goal: addressing the interpretability flaw in dense activation-based encoding models.
>
> ymzV suggests we draw on prior work, but we know of no prior study using *sparsified* LLMs for brain encoding. Developing a sparsified Gemma Scope-based model necessitates many non-trivial decisions due to the complexity & scale of LLM activations. Whereas GloVe-based sparse embedding has 1000 dimensions, even the smallest Gemma models have *millions*, with atom-level distinctions as fine-grained as “phone numbers” vs. “scientific numbers”, greatly increasing interpretive & computational burden. Our revision can add a limited demonstration that our core results generalize to LLM-derived features, but full methodological treatment of sparsified LLMs is beyond the scope of the paper & better suited for a dedicated followup.

---

### Note · Authors · 2025-08-14

At the start of the review process, the panel was split. Two reviewers (iQoW and ZoiN) were enthusiastic from the outset, recommending acceptance. The other two raised two major criticisms: lack of novelty (buH2 and ymzV) and insufficient quantitative analysis (ymzV).


- To address buH2 and ymzV’s concerns about novelty:

  - We clarified that our contribution does not lie in the novelty of the individual techniques—dictionary learning and voxelwise regression, but in identifying and overcoming a previously overlooked limitation of brain encoding models using ANN features. Many recent papers, including several published at NeurIPS, assume that learned weight vectors in ANN activation spaces are directly interpretable. We formally demonstrate that this assumption is flawed due to superposition, and propose the Sparse Concept Encoding Model as a practical solution that enables principled neuroscientific interpretation. This limitation and its solution are broadly relevant to ongoing work in the community. In revision, we committed to specific changes, including reframing to clearly state our scientific goal and moving the theoretical discussions into the main text. See rebuttal to buH2 for full details.

ymzV’s concern about quantitative analysis had two main parts: inadequate number of subjects and insufficient quantitative descriptions alongside visualizations.

- To address these concerns:

  - We conducted full set of analyses across all seven subjects and reported complete statistics. While we initially explained our framework prioritizes within-subject analysis, which was the intended scope for this conference paper, we provided the full analysis after ymzV emphasized that cross-subject evaluation was still necessary.


  - We provided the detailed quantitative data requested. We had agreed in our first reply that adding these metrics in revision would strengthen the paper. When ymzV reiterated the concern, noting the absence of explicit numerical values, we responded in our latest comment with quantitative results, including inter-subject consistency, IoU measures, and concept-level correlation statistics.


Outcome:

Following these changes, buH2 recommended acceptance. ymzV did not respond further after our comment with the full additional analysis, but every methodological point they listed was addressed and no substantive concerns remain unresolved. The paper now combines a clearly articulated conceptual advance with robust empirical support.

---

### Decision · Program_Chairs · 2025-09-17

**Decision:**

Accept (poster)

**Comment:**

This paper proposes to use sparse positive decompositions of dense learned embeddings to achieve more interpretable mappings from learned representations to neural data. The method is sensible and clearly presented, and the qualitative results are suggestive.

 While most the reviewers were satisfied with the authors' responses, I find myself still somewhat sympathetic to several of reviewer ymzV's concerns; it is not entirely clear to me from the qualitative analyses presented that this model is better than those used in prior work (even the simple clustering method originally used by Huth et al., 2016), and the authors don't seem to make a compelling case for this point (e.g. showing that the dense embeddings *do not* reveal the same structure in the neural data). The closest they come is the mathematical argument, but 1) I don't find the mathematical argument for non-identifiability to be particularly novel (it basically just says that the null space of a non-invertible matrix is non-trivial, which is just a fundamental fact about linear algebra), and 2) showing that another method can fail to recover full information in some cases is not a sufficient argument for why a new method is better. Sparse factorization can *also* be non-identifiable in some cases. I'd encourage the authors to take these issues into consideration in revising the paper.

On the whole, I find this paper to be just over the threshold of interestingness to accept, despite its limitations and framing issues.